

# Recent trends in climate variability at the local scale using 40 years of observations: the case of the Paris region of France

Justine Ringard[1], Marjolaine Chiriaco[1], Sophie Bastin[1], Florence Habets[2-3]

[1]LATMOS/IPSL, UVSQ Université Paris-Saclay, Sorbonne Université, CNRS, Guyancourt France
[2]METIS/IPSL, Sorbonne Université, CNRS, Paris France
[3]Laboratoire de Géologie, Ecole normale supérieure, PSL Research University, CNRS UMR 8538, 24 rue Lhomond, 75005 Paris, France

*Correspondence to*: Justine Ringard (justine.ringard@latmos.ipsl.fr)

**Abstract.** For several years, global warming has been unequivocal, leading to climate change at global, regional
and local scales. A good understanding of climate characteristics and local variability is important for adaptation
and response. Indeed, the contribution of local processes and their understanding in the context of warming are still
very little studied and poorly represented in climate models. Improving the knowledge of surface-atmosphere
feedback effects at local scales is therefore important for future projections. Using observed data in the Paris region
from 1979 to 2017, this study characterizes the changes observed over the last 40 years for six climatic parameters
(e.g., mean, maximum and minimum air temperature at 2 metres, 2 metres relative and specific humidities and
precipitation) at the annual and seasonal scales and in summer, regardless of large-scale circulation, with an
attribution of which part of the change is linked to large scale circulation or thermordynamic. The results show that
some trends differ from the ones observed at the regional or global scale. Indeed, in the Paris region, the maximum
temperature increases faster than does the minimum temperature. The most significant trends are observed in spring
and in summer, with a strong increase in temperature and a very strong decrease in relative humidity, while specific
humidity and precipitation show no significant trends. The summer trends can be explained more precisely using
large-scale circulation, especially regarding the evolution of the precipitation and specific humidity. The analysis
indicates the important role of surface-atmosphere feedback in local variability and that this feedback is amplified
or inhibited in a context of global warming, especially in an urban environment.

# 1 Introduction

The climate system warming is unequivocal, and since the 1950s, many observed changes have been unprecedented (GIEC, 2014). At the global scale, this warming has shown a trend of approximately 0.12 °C per decade since 1951 (GIEC, 2014), with a greater change in daily minimum temperatures (Tmin) than daily maximum (Tmax) ones (Donat and Alexander, 2012).



Donat et al., 2013 analysed 27 indices of temperature and precipitation recommended by the Expert Team on Climate Change Detection and Indices (ETCCDI; Karl et al., 1999; Peterson et al., 2001; Peterson, 2005) over the last century in different regions of the globe and concluded that more significant change has occurred in Tmin extremes, especially in recent decades and that most regions have experienced an increase in precipitation intensities and frequencies.

Previous studies suggested that a change in average temperatures largely explains the change in observed extreme temperatures (Rhines and Huybers, 2013; Tingley and Huybers, 2013; McKinnon et al., 2016). Donat and Alexander, 2012 studied the maximum and minimum daily temperature variations on the globe, comparing the probability density functions (PDF) of the variables between 1951-1980 and 1981-2010. Comparisons showed that both maximum and minimum daily temperatures have shifted to higher values over the last 30 years in all regions of the world. However, changes in variance and asymmetry of

distributions between the two periods are spatially heterogeneous. This result is also confirmed by Kodra and Ganguly, 2015 and McKinnon et al., 2016, who show that the change in distribution depends on the region and the season. At these regional and local scales, changes in temperature distribution may be more complex than at the global scale (Huntingford et al., 2013). We still do not know how to determine if extreme climate change is simply a result of an increase in the mean or higher order changes, such as variance, skewness and / or kurtosis (Lewis and King, 2017), because, according to Donat et al., 2013, regional

changes affect the higher order of the statistical moments of a distribution.

Western Europe is warming much faster than expected, with an increase of temperatures of 0.5°C per decade over the past 30 years (Van Oldenborgh et al., 2009) and summer temperatures increasing more rapidly since the mid-1990s (Dong et al., 2017). Climate variability in Western Europe is influenced by both large-scale dynamics and regional / local physical processes. Global warming causes dynamic and thermodynamic changes that modify the extreme event probabilities (Horton et al., 2015).

In a pioneering study, Reinhold and Pierrehumbert, 1982 suggested that observed changes at the surface may result from a progressive change in the frequencies of occurrence of different climatic regimes, but the change in atmospheric circulation controlled by large-scale dynamics, linked to global warming, shows sources of uncertainty, notably in projections (Cohen et al., 2014; Shepherd, 2014). Several studies show that in winter, changes in atmospheric circulation are the main drivers of surface weather trends in Europe (Van Oldenborgh et al., 2009; Vautard and Yiou, 2009; Yiou et al., 2018). In summer, this

is not always the case because of the strong interaction between the temperature and the water cycle (Van Oldenborgh et al., 2009; Vautard and Yiou, 2009). Cattiaux et al., 2012 have nonetheless suggested that the observed increase in interannual variability could be partly explained by atmospheric dynamics. However, the influence of other processes (such as heat fluxes or evapotranspiration driven by soil moisture), which are more important in summer and spring, suggests that recent and future warming in Europe is incompatible with changes in atmospheric circulation alone, and surface-atmosphere processes are

mainly responsible for increasing temperature variability, especially summer temperatures (Zampieri et al., 2009). In addition, various studies, such as those of Chiriaco et al., 2014 and Miralles et al., 2014, show that the development of summer heat waves, their frequency and their intensity result from a combination of large-scale specific atmospheric circulation and specific surface-atmosphere interactions. Some thermodynamic changes related to global warming are well understood and observed, such as humidity-temperature positive feedback (Seneviratne et al., 2010; Fischer et al., 2012; Miralles et al., 2014; Cattiaux



et al., 2015; Vogel et al., 2017). However, other surface-atmosphere feedbacks are still very little studied in the context of global warming and are poorly represented in the models (Vautard et al., 2018). For example, in Western Europe, the number of anticyclonic situations increases, but the amount of rainfall does not decrease, which indicates an increase in the amount of rain per event and not an increase in the number of events (Vautard and Yiou, 2009). However, Rasmussen et al., 2017

determined that downstream of the Rockies in the US Great Plains during summer, both the magnitude of the Convective Available Potential Energy (CAPE) and the Convective Inhibition (CIN) increase in a warmer future climate. This means that the triggering of precipitation will be more difficult, reducing the frequency of weak to moderate precipitation, but the intensity of precipitation when it occurs will increase in a future climate, thereby modifying the spatial and temporal occurrence of precipitations. Bastin et al., 2019 highlighted the importance of surface-atmosphere processes, particularly at the local scale,

because triggering thresholds are a function of local moisture sources. The contribution of local processes and their understanding in a warming context is therefore an important factor in improving future projections.

The purpose of this article is to characterize, at the local scale, trends and changes in temperature distributions (T2m, Tmax and Tmin), relative and specific humidities and precipitation over the last four decades in the Paris area, and to deduce the possible modifications of the surface-atmosphere feedbacks. The study focuses on the Paris region for the following several

reasons: i) as a densely populated area (11 million inhabitants), it is sensitive to extreme events such as floods (2001, 2016, 2018) and heat waves (2003, 2006, 2018), which are projected to become more common in the future; ii) the Paris area is conditioned by an urban microclimate (urban heat island), which intensifies extremes, while very few tools related to climate modelling are adequate to study the impact of urban areas; and iii) the supersite SIRTA (Site Instrumental de Recherche par Télédétection Atmosphérique / Instrumental Research Site of Atmospheric Teledetection in Palaiseau (Haeffelin et al., 2005),

cyan in Fig. 1), which monitors long-term multiple variables with high-level measurements, is located in the area. These specific observations are not used in the current paper.

The current paper examines changes in temperature distributions (T2m, Tmax and Tmin), humidity (relative and specific) and precipitation at the following different time scales: annually, seasonally (winter DJF, spring MAM, summer JJA and autumn SON) and especially during the summer season, according to the large-scale circulations; the paper also compares these

changes between a past period (1979-2002) and current period (2003-2017). The observations used and their comparisons to reanalysis products are presented in Section 2 in order to check the ability of the commonly used atmospheric reanalyses to detect local scale trends. Section 3 describes the methodology used to calculate trends and describe PDFs, as well as the method used to define continuous time weather regimes. In Section 4, trends and changes in distributions of the different parameters are analysed on an annual and seasonal scale. In Section 5, a focus is made on the summer season to analyse trends and

distributions for the four main large-scale circulations. Section 6 discusses the results and their possible relationships according to the nature of the change (dynamic, thermodynamic or anthropic).



## 2 Data

### 2.1 Observations

This study analyses climate trends at the local scale, in the region of Paris, France, from 1979 to 2017 using five Météo France (MF) daily observation stations (in yellow in Fig. 1). The choice of these stations is motivated by i) their good temporal
sampling (few measurement gaps), ii) their temporal availability, since at least 1979, and iii) the fact that all considered variables are measured at the station. We performed a sensitivity analysis to compare the variability between each station as well as their differences with the ensemble of the five stations' mean. For all variables, the five stations show a very strong correlation between them (not shown). The Montsouris (Trappes) station, located downtown (outside) Paris, has slightly warmer (colder) temperatures but similar variability to the other stations. Regarding the relative humidity, Montsouris has
slightly drier conditions and Trappes slightly wetter conditions. Note that the other three stations show a correlation and variability almost identical to the average of the stations. This is why the five stations are averaged together to obtain a single daily observation series corresponding to the "Paris region". For the sake of comparison (Section 2.2 and Appendix A), for precipitation, the average of the four stations located within the SAFRAN grid (in orange in Fig. 1) is computed.

### 2.2 Comparison of the local observation with the ERA-Interim reanalyses and SAFRAN analysis

Although the main data sources in this study come from direct observation, it is interesting to test the ability of well-known reanalyses to represent the fine-scale behaviour. To do so, we used the reanalysis from the European Centre for Medium-Range Weather Forecasts (ECMWF) ERA-Interim (Simmons et al., 2014), as well as the high resolution meteorological analysis SAFRAN (Quintana-Seguí et al., 2008), for precipitations. ERA-I shows a general pattern of underestimation of temperatures (T2m, Tmax and Tmin) relative to observations, which is more marked seasonally, especially in spring and
summer (Fig. A1b and A1e). In addition, ERA-I also shows a strong overestimation of relative humidity annually and seasonally, whereas the specific humidity is rather well estimated by ERA-I. For precipitation, SAFRAN shows rather satisfactory results in terms of bias despite the high daily variability (Fig. A1b and A1e). However, the statistical analyses carried out on the monthly accumulations show very good results, confirming that SAFRAN is well adapted to inform the precipitation at local scale, at least for this area. The detailed results obtained from the two datasets are presented in Appendix
A. The reasons for the discrepancies between direct observation and ERA-I / SAFRAN are out of the scope of this paper, but the presence of significant bias at this local scale motivates the use of observations and not reanalysis for the current issues.

### 2.3 Local climate

The temporal evolution of the six daily variables, namely, the daily temperature at 2 metres (T2m), the daily maximum temperature at 2 metres (Tmax), the daily minimum temperature at 2 metres (Tmin), the relative humidity (RH), the specific
humidity ($q$) and the precipitation (PRCP), on an annual basis and for seasonal scales are presented in Fig. 2. The local climate is characterized by cold and humid winters in contrast to warm and increasingly drier summers. The seasonal averages of T2m



and Tmax are similar in spring and autumn; however, autumn has warmer Tmin and wetter conditions than does spring. The relative humidity is the only variable for which the decrease tendency, especially in the spring-summer, clearly appears. Regarding precipitation, the Paris region shows no preferential season when considering the total amount.

## 3 Methodology

### 3.1 Mann-Kendall Trend Test

Trends were calculated using the Mann-Kendall test (Mann, 1945; Kendall, 1955). This test detects the presence of a monotonic tendency in a chronological series of a variable. It is a non-parametric method; that is, it makes no assumptions about the underlying distribution of the data, and its rank-based measure is not influenced by extreme values. This method mainly gives three types of information.

- The Kendall Tau, or Kendall rank correlation coefficient, measures the monotony of the slope. It is positive when the trend increases and negative when the trend decreases. Kendall's Tau varies between -1 and 1: the closer Kendall's Tau is to 1 (-1), the more positive (negative) correlation of the variable with time is certain.
- The significance, which represents the threshold for which the hypothesis that there is no trend is accepted. The trend is statistically significant when the p-value is less than 0.05.
- The slope of Sen, which estimates the overall slope of the time series. This slope corresponds to the median of all the slopes calculated between each pair of points in the series.

### 3.2 Anomaly and Normalization

The data of an $X$ variable are exploited as anomalies $X'$ with respect to climatology $\bar{X}$. Here, we take as climatology the whole period of study, the normal of a day $d$ of the year $y$, with $d \in$ [1 à 365] as the average of this day over the period 1979-2017 (Eq. (1)) as follows:

$$\bar{X}(d) = \frac{1}{N} \sum_{y=1979}^{2017} X(d) \tag{1}$$

with $N$ number of years. To obtain a non-noisy signal, the climatology $\bar{X}$ is smoothed by a LOWESS, i.e., a "LOcally Weighted Scatterplot Smoother", with a spar = 0.3. The spar represents the fraction of data used to smooth the series; it is between 0 and 1. Here, we retain a spar of 0.3 to sufficiently smooth the series by attenuating the residual component, i.e., the noise, while maintaining the appearance of the trend. Once the climatology is obtained, we calculate the daily anomaly (Eq. (2)) as follows:

$$X'(d) = X(d) - \bar{X}(d) \tag{2}$$




Monthly or seasonal anomalies are directly obtained by averaging $\bar{X}(d)$ over months or seasons. Finally, the anomalies $X'(d)$ are normalized over the period 1979-2017 according to the temporal scale studied t, where t $\epsilon$ 1,..., $Nt$ (year or season) as follows:

$$\widetilde{X'}_t = \frac{X'_t - \mu_{X'}}{\sigma_{X'}} \qquad (3)$$

With

$$\mu_{X'} = \frac{1}{N_t} \sum_t X'_t \qquad (4)$$

$$\sigma^2_{X'} = \frac{1}{N_t-1} \sum_t (X'_t - \mu_{X'})^2 \qquad (5)$$

We normalized the anomalies with respect to the entire 1979-2017 studied period because, according to Huntingford et al.,
2013 and Sippel et al., 2015, when anomaly normalization is performed relative to a reference period, then standardization tends to increase the variability and extremes, especially at the global scale.

### 3.3 Statistical characteristics of the PDFs

Global warming is accompanied by changes in the distributions of climate variables. Several kinds of changes can occur, including i) a "change in the mean", where there is an increase or decrease of the probability distribution by shifting to the
right or to the left, ii) a "change in symmetry", for example, the distribution spreads to the right so that the lowest tail of distribution would remain approximately at historical intensities and the distribution of the highest extremes would increase, and iii) a "change in variability", where there is a symmetrical widening, i.e., a flattening of the distribution, leading to an increase in both cold and warm extremes in the case of temperature (Donat and Alexander, 2012; Field et al., 2012; Lewis and King, 2017).

Normalized anomaly distributions for two periods (1979-2002, 2003-2017) are analysed and the Probability Density Functions (PDF) are calculated. For each PDF, we calculate the symmetry coefficient or skewness "S", as well as the shape coefficient or kurtosis "K". The symmetry coefficient, which is the moment of order 3, is without unity. If the distribution is symmetrical, this coefficient is equal to zero; if the distribution spreads to the left (right), it is negative (positive). The shape coefficient, which is the moment of order 4, measures the flattening of the distribution. The kurtosis of any normal distribution is 3. The
larger the value, the sharper the distribution. Conversely, the smaller the coefficient, the flatter the distribution, which leads to greater variability.

The choice of separation between these two periods 1979-2002 and 2003-2017 is mainly motivated by the fact that over the period 2003-2017, observations of various meteorological parameters are available at the supersite SIRTA (see Fig. 1) and have been reanalysed to produce the SIRTA-ReOBS dataset at an hourly time scale (Chiriaco et al., 2018). This dataset is not
used in this study, but it will be used in a forthcoming paper focused on understanding the processes responsible for the changes detected in the current paper.



### 3.4 Climate indices

We used the climate indices recommended by the joint CCl (WMO Commission for Climatology) / CLIVAR (World Climate Research Programme Project for Climate Variability and Predictability) / JCOMM (Joint Technical Commission for Oceanography and Marine Meteorology) Expert Team on Climate Change Detection and Indices (ETCCDI) (Karl et al., 1999;

Peterson et al., 2001; Peterson, 2005; Zhang et al., 2011) calculated from Tmax, Tmin and PRCP (Table 1). With regard to precipitation, the indices are generally calculated according to a threshold of 1 mm; this threshold differentiates a rainy day from a non-rainy day. In this study, we modified this threshold to 0.2 mm. This choice is motivated by the World Meteorological Organization (WMO), which recommends an accuracy of 0.2 mm for rain gauges (WMO, 2014), considering that minimal rainfall for a rainy day is 0.2 mm d$^{-1}$. Finally, when the indices use percentiles, they are calculated on the annual

series when looking at annual trends and calculated over the season when looking at seasonal trends.

### 3.5 Weather Regimes

In winter and summer, climate variability in Western Europe is controlled by different dynamic states called weather regimes (Cassou et al., 2005, 2011). These regimes are interpreted as quasi-stationary states of daily atmospheric circulation that can persist from a few days to a few weeks. Michelangeli et al., 1995 show that four regimes are relevant for the study of climate

variability in the North Atlantic-European basin (NAE). These regimes are defined according to the geopotential height at 500 hPa or the sea level pressure (SLP) by the k-means method. Thus, each day is associated with a preferential regime (Legras and Ghil, 1985; Vautard, 1990; Yiou et al., 2008). Weather regime analysis allows observing climate trends at constant air mass; that is, large-scale circulation is fixed, and thus the variability detected is rather explained by smaller scale processes. This study uses a regime classification, calculated from the SLP over a reference period 1970-2010 and available at the

following link https://a2c2.lsce.ipsl.fr/index.php/deliverables (for more details see Cattiaux, 2010; Yiou et al., 2011, 2018). Such classification is efficient for stable seasons such as winter and summer, and less for spring and fall, which are transition seasons and therefore more subject to rapid, large-scale changes. We mainly focused on summer because of the strong local variability related to the thermodynamical processes that affect the summer season and whose changes are more marked in summer than in winter.

In summer, there are four preferential regimes, as detailed below (Fig. B1 in the Appendix B represents the anomalies of SPL associated with these four summer regimes).

- The NAO- phase (Fig. B1) is characterized by a weakening of the Icelandic Low. The jet stream is pushed back to the south on its arrival in Western Europe, causing cold conditions over most of Europe. In the Paris area, this regime is marked by cooler and wetter conditions.

- The Atlantic Ridge phase is characterized by high pressures over the Atlantic Ocean and low pressures over the northwest of Europe, favouring cold conditions via the reinforcement of a polar flux. On the other hand, it inflates





the Azores Anticyclone in its subtropical part and thus warms the rest of Europe. In the Paris area, this regime is marked by cool temperature and slightly humid conditions.

- The Blocking phase is characterized by a strong anticyclone over the British Isles, which blocks the inflow of maritime air and allows warm conditions to develop in Western Europe. Southeast Europe is rather cold. In the Paris region, this regime favours hot and dry temperature conditions.

- The Atlantic Low phase slows down the polar flow in favour of a southerly flow favourable to warm conditions over all of Western Europe. In the Paris region, this regime favours warmer and drier conditions than other regimes.

Thus, each summer day of our study is associated with one of the four weather regimes above, and we can separate at first order the evolutions of the parameters due to circulation changes to those due to local changes.

## 4 General results

In this section, the observed trends for several variables and climate extremes indices at the annual scale since 1979 are presented. Then, in a second step, each variable and each climate extreme index is studied at the seasonal scale.

### 4.1 Annual Trends

At the annual scale, Mann-Kendall trends from observations (Fig. 3a) show a significant increase in T2m of approximately 1.6°C since 1979 (0.4°C decade$^{-1}$), 1.9°C for Tmax (0.47°C decade$^{-1}$) and 1.5°C for Tmin (0.37°C decade$^{-1}$). In addition, as the Tmin Kendall Tau is higher than that of Tmax, this means that although Tmin warms up less quickly than the Tmax, its increase is more monotonic. The relative humidity decreases significantly (4.3 %, i.e., 1.24 % decade$^{-1}$) from 79.2 % (origin of Sen slope in 1979) to 74.5 % in 2017, and it appears to be guided by the temperature trend, as no significant trend is detected for specific humidity. For precipitation, despite an observed decline, there is no significant trend.

Figure 3b shows the trends, on an annual scale, of climate indices calculated from Tmax, Tmin and PRCP (see Table 1 for definition).

- For the warm part of the distribution, warm Tmin (Tn90p) increases significantly, and the number of summer days (SU) shifts from approximately 37.7 days in 1979 to 50.3 days in 2017.

- For the cold part of the distribution, Tx10p and Tn10p decrease significantly, as well as the number of frost days (FD) from approximately 44 days in 1979 to 26 days in 2017.

- For precipitation, only the maximum number of consecutive wet days decreases significantly, with a maximum period of consecutive rainy days equal to 12 days on average in 1979 and 8 days in 2017.

Hence, on an annual scale, in the Paris region, the changes of the last four decades are mainly on the relative humidity, which presents a strong decrease, and on the temperatures (average, maximum and minimum), with a shift of the distribution towards warmer temperatures leading to more warm days, fewer cold days, and higher minimum and maximum temperatures, which is a rather typical trend, although Tmax presents a stronger positive trend than Tmin. No significant trend can be detected for





precipitation because the variability is too great, except for the decrease of the maximum number of consecutive rainy days. This result is opposite to the one reported by Zolina et al., 2010, who found that wet spells increase over 60 years in Europe by approximately 15 to 20 %. However, the period of study differs substantially as they carry out their analysis over the 1950-2008 period, and their threshold between a rainy and non-rainy day is 1 mm versus 0.2 mm for the current study. The analysis

of precipitation can be sensitive to these differences and to local effects. It is expected that the decrease in relative humidity observed in the Paris area affects some indices of precipitation, especially indices concerning occurrence (Bastin et al., 2019).

### 4.2 Seasonal Trends

Our study area is marked by a high seasonal cycle (Fig. 2). For each variable, we apply our analysis for the four seasons as follows: the winter season from December to February (DJF), the spring season from March to May (MAM), the summer

season from June to August (JJA) and the autumn season from September to November (SON).

### 4.2.1 Temperatures

For all seasons except winter, T2m increases significantly (Fig. 4a), approximately 2.1°C (0.52°C decade$^{-1}$) in spring and 1.8°C (0.46°C decade$^{-1}$) in summer, with a strong positive monotonic relationship (Kendall's tau). Warming is also significant for Tmax (Fig. 4b) and Tmin (Fig. 4c) at all seasons except for DJF. Tmax increases strongly in MAM (2.9°C, i.e., 0.73°C decade$^{-1}$)

and JJA (2.1°C, i.e., 0.52°C decade$^{-1}$), while the Tmin increase is slightly weaker (1.6°C, i.e., 0.41°C decade$^{-1}$ in MAM, 1.8°C, i.e., 0.46°C decade$^{-1}$ in JJA). However, the Kendall Tau of Tmin is greater than 0.4 in JJA; this is the largest Tau for all temperatures and all seasons, reflecting a constant increase in Tmin in JJA since 1979.

In terms of PDF and extremes, DJF shows little change in the mean of the PDF (Fig. 4d), but the number of very cold anomalies of T2m ($<-3\sigma$) decreases. The same results are observed for Tmax and Tmin (not shown), but there are no trends in temperature

climate indices (Fig. 5a).

In MAM (Fig. 4e), the average of the T2m anomalies over the current period increases, marked by a shift of the PDF to the right, which means more warm anomalies. The number of days where Tmax is lower than the 10$^{th}$ percentile (Tx10p) decreases (Fig. 5b), consistent with the strong increase in Tmax for this season. On average, Tmax warms up very strongly (2.9°C in 39 years, from approximately 13.9°c to 16.7°C, Fig. 4b), with constant behaviour (strong Kendall tau, Fig. 4b). However, there

is no change in the cold anomalies tail of the distribution of Tmax (not shown). This indicates that the presence of very cold events persists in spring, but with a decline in frequency (Tx10p, Fig. 5b). Figure 5b also shows that the percentage of days when the minimum temperature is greater than the 90th percentile (Tn90p) increases in spring.

In JJA (Fig. 4f), the average T2m anomalies increase (PDF less flattened with K>3 and shifted to the right with S=0.6), as well as very warm anomalies greater than 2$\sigma$. The same characteristics of PDF evolution are observed on Tmax and Tmin (not

shown). The temperature indices show strong significant trends (Fig. 5c). The cold indices (Tx10p and Tn10p) decrease continuously, whereas warm indices (Tx90p, Tn90p, TR) increase. In summer, high values of Tmin (higher than the 90$^{th}$ percentile) were reached for 3.8 % of the days in the past compared with 13.6 % now; at the same time, the lowest temperatures





($10^{th}$ percentile) were reached for 15.7 % of the summer days in the past and only 4.4 % of present days. These trends are linked to the strong increase in Tmax and Tmin observed in JJA and in particular the right shift of the PDF.

In SON (Fig. 4g), the same as for the other seasons, the average T2m anomalies increase, cold anomalies are less cold and warm anomalies are more likely to occur. Tmax and Tmin show the same characteristics. Significant trends are observed for the coldest temperature indices (Fig. 5d). The number of days where the Tmin is less than the $10^{th}$ percentile (Tn10p) and the number of days where the Tmin is below 0°C (FD) decreases significantly. These results come from the increase in Tmin (1.5°C), which in autumn is larger than for the Tmax (1.1°C).

In summary, the largest temperature changes appear in MAM and JJA. Spring shows a strong increase in temperatures, but climate indices show fewer changes due to variability, allowing the presence of punctually cold Tmax and Tmin. In summer, the temperatures increase strongly, as do the very warm anomalies higher than 2σ; warm (cold) extremes are more (less) frequent. In the autumn, cold extremes decrease due to the stronger increase of Tmin than Tmax.

### 4.2.2 Humidity

The relative humidity (Fig. 6a) decreases significantly in all seasons except DJF. This is due to the fact the specific humidity increases are less than what could be expected by Clausius-Clapeyron, according to the increase of the temperature. Indeed, specific humidity shows no significant trends and even shows a slope of zero in JJA (Fig. 6b). The strong monotonic decrease of RH is approximately 7.7 % (1.92 % decade$^{-1}$) in MAM and 8 % (1.99 % decade$^{-1}$) in JJA. For JJA, RH shows an average value of 72.3 % in 1979 and decreases to 64.7 % on average in 2017. This strong decrease in relative humidity is observed on PDFs (Fig. 6c-f). For all seasons, the average of the anomalies decreases (current PDF shifted to the left). However, this shift is more marked in MAM (Fig. 6d) and JJA (Fig. 6e). In addition, DJF shows little change in the extremes (Fig. 6c), while in MAM and JJA, the number of moist anomalies decreases, and the number of dry anomalies increases. Finally, in SON, the number of very humid anomalies decreases, and the number of very dry anomalies increases (Fig. 6f).

In summary, in spring, summer and autumn, the evolution of RH distribution leads to a decrease in the frequency of humid anomalies (very humid anomalies in autumn) and an increase in the frequency of dry anomalies (very dry anomalies in autumn). This decrease appears to be guided by the temperature trend, as the amount of water in the atmosphere near the surface, ie, specific humidity, remains almost unchanged at all seasons.

### 4.2.3 Precipitation

At the seasonal scale, the trends in rainfall are not significant (Fig. 7a). Figure 7b-e shows the PDFs of observed daily intensities only for rainy days (>0.2 mm day$^{-1}$) for the past period (1979-2002) and the current period (2003-2017). In DJF (Fig. 7b), the frequency of daily intensity decreases over the current period, also observed on climatic indices with a decrease in R90pTOT (Fig. 8a). In addition, the maximum number of consecutive wet days (CWD) decreases (Fig. 8a) from approximately 10.1 to 6.8 days. In MAM, the extreme intensities of precipitation are slightly more frequent over the current period (Fig. 7c). Furthermore, Fig. 8b shows a decrease in the percentage of rainy days (% rainy), a decrease in the maximum number of



consecutive wet days (CWD) and an increase in the maximum number of consecutive dry days (CDD). The spring shows, on average, 48.5 % of rainy days in 1979 versus 36.3 % in 2017, and the average maximum periods of consecutive dry days evolve from 8.7 to 15.4 days. In spring, the weather is drier with fewer rainy days, but slightly more extremes. This is consistent with the decrease in relative humidity that affects the triggering of precipitation (Rasmussen et al., 2017; Bastin et al., 2019).

In JJA (Fig. 7d), the frequency of mean intensities (PRCP between 12 and 20 mm day$^{-1}$) increases and the frequency of extreme intensities decreases. However, in JJA, despite all rainfall indices showing an increase, none is significant (Fig. 8c). In SON (Fig. 7e), the frequency of daily intensity decreases over the current period, a result also observed with the significant decrease of the SDII (Fig. 8d), i.e., a decrease in the daily mean intensity.

In summary, the high variability of precipitation does not allow the detection of significant trends for most climate indices.
Nevertheless, the indices emphasize some results: extremes of precipitation occur less frequently in DJF, MAM becomes drier but heavy precipitation is stronger, JJA shows no significant trends and SON is marked by a decrease of the mean daily intensity.

The analysis shows that unexpected changes are occurring in summer at first order: precipitations exhibit an increasing trend (not significant), while it is the only season for which the specific humidity does not increase. To further study this season, it
is necessary to understand what happens for each of the main atmospheric circulations. In the following section of this study, we focus on the summer season and we perform our trend analyses independent of large-scale circulations in order to characterize the changes coming only from thermodynamical processes.

**5 Focus on the summer season**

Changes in temperature, relative humidity, and precipitation, both in trends and distribution patterns, are more pronounced in
spring and summer. The intra-seasonal changes are identified based on a classification of each summer day based on weather regimes, which allows characterizing both the changes associated with large-scale circulation (in frequency) and the changes within each weather regime. Van Oldenborgh et al., 2009 and Vautard and Yiou, 2009 found that changes in atmospheric circulation are not the main drivers of surface weather patterns in summer, unlike in winter. Indeed, local physical processes play a major role in summer variability. In the rest of this study, we focus on summer rather than spring because i) the large-
scale dynamics are more stable, which allows the definition of weather regimes and then the separation of the variability due to the large scale from that due to more local processes, ii) the relative humidity decreases significantly in summer and spring, but the increase (nonsignificant) of specific humidity is particularly reduced in summer, and iii) the evolution of precipitation indices in MAM is consistent with the decrease of relative humidity, but not those in summer. For each regime, the percentages of frequency are computed for the past period (1979-2002) and the current period (2003-2017). The frequency of NAO- and
Atlantic Low regimes increased by 9.1 % and 3 %, respectively; and the frequency of Atlantic Ridge and Blocking regimes decreased by 5.9 % and 6.2 %, respectively. Using the weather regimes, we can write the temperature T2m (or precipitation PRCP) as the sum, for the four regimes, of the occurrence of regime i * the mean value of temperature (or the daily mean



intensity of precipitation RR) in this regime. Then, between the two periods, we can calculate the dynamical and thermodynamical contributions of the change of the variable considered ($\Delta T$ for temperature or $\Delta PRCP$ for precipitation) adapted from Cassano et al., 2007 and Screen, 2017 according to the following equations:

$$\Delta T = \sum_{i=1}^{4}(\Delta f_i \overline{T_i} + \Delta T_i \overline{f_i} + \Delta f_i \Delta T_i) \tag{6}$$

$$\Delta PRCP = \sum_{i=1}^{4} \Delta PRCP_i \tag{7}$$

with    $\Delta PRCP_i = \Delta f_i \overline{RR_i} + \Delta RR_i \overline{f_i} + \Delta f_i \Delta RR_i$ (8)

For example, with precipitation for a weather regime $i$, $\Delta f_i$ and $\overline{f_i}$ are respectively the difference in the frequency of occurrence of the regime between the two periods and the mean value of frequency of occurrence in the past period; $\Delta RR_i$ and $\overline{RR_i}$ are, respectively, the difference in the daily mean intensity of the precipitation between the two periods and the daily mean intensity of the precipitation in the past period. Then, $(\Delta f_i * \overline{RR_i})$ is considered the dynamical term (change of precipitation due to

dynamical change), $(\Delta RR_i * \overline{f_i})$ the thermodynamical term (change of precipitation due to thermodynamical change) and $(\Delta f_i * \Delta RR_i)$ is the residue. Using Eq. (7), in summer, the precipitation changes observed (Table 2) are explained at 67.6 % by the thermodynamical contribution and 32.4 % by the dynamics of occurrence, whereas at the weather regime time scale, the dynamics of occurrence are greater than the thermodynamics, contributing between 47.9 % and 88.7 % to the precipitation change observed.

For climatic indices based on percentiles, we computed one value of the percentiles using the distribution of the entire summer season and the whole period but not a value for each weather regime. In this way, it is possible to characterize the evolution of each index inside a regime but also to compare the indices between regimes.

### 5.1 NAO-

The NAO- regime is characterized by a weakening of the Icelandic Low. Conditions are generally cooler over most of Europe.

Since 1979, for NAO-, T2m increases significantly by 1.9°C (0.49°C decade[-1]; Fig. 9), Tmax by 2°C (0.52°C decade[-1]; not shown) and Tmin by 1.8°C (0.45°C decade[-1]; not shown). This weather regime shows the largest increase in T2m (and Tmin) compared to other summer regimes. Climatic indices' trends are not significant (Fig. 10a), but we observe fewer days below the 10[th] percentile for Tmin and Tmax and more hot days, while it is a weather regime associated with fresh conditions. Specific humidity (Fig. 9) shows little difference in the distribution, although most humidity is advected from the Atlantic Ocean during

this regime. Such an evolution, associated with a temperature increase, is consistent with a decrease in the relative humidity (Fig. 9), but this decrease is weaker than for the other summer regimes. Rainfall increases, but not significantly (Fig. 9 and PRCPTOT Fig. 10a). NAO- is the only weather regime that shows an increase in PRCPTOT (Fig. 10); the intensity of this



increase (~ 8 mm decade$^{-1}$) corresponds to the total increase observed in JJA (Fig. 8). Two reasons could explain this trend: precipitation increases during this regime (occurrence or intensity by event or both), or this trend is related to the increase in the number of days in NAO-. By applying Eq. (8) to determine the origin of change in precipitation between the two periods, the results presented in Table 2 show a contribution of the dynamical term, which is preponderant over the thermodynamical

term, with an increase in the frequency of occurrence of days in NAO- (+9.1 %, i.e., approximately 8 days), explaining 87.7 % of precipitation change observed in this regime. Furthermore 1) the mean and median daily precipitation values are the same between the two periods (Fig. 9) and 2) the mean intensity of rainy days (SDII Fig. 10a) and the percentage of rainy days (% rainy Fig. 10a) show almost zero trends. All of these reasons confirm that the increase in PRCPTOT in NAO- (hence in JJA) is more related to an increase in the occurrence of days in NAO-.

**5.2 Atlantic Ridge**

The Atlantic Ridge regime is characterized by high pressures over the Atlantic Ocean and low pressures over northwestern Europe, favouring cold conditions through the enhancement of polar flux towards Western Europe. On the other hand, it inflates the Azores Anticyclone in its subtropical part and thus warms the rest of Europe. Under this regime, the temperatures over the Paris area increase significantly for T2m (1.7°C, i.e., 0.43°C decade$^{-1}$; Fig. 9b) and especially for Tmin (1.8°C, ie,

0.45°C decade$^{-1}$; not shown). Warm and very warm anomalies are more frequent, but most striking is the change of shape of the violin, with a crushing of the bottom of the distribution and a stretching of its top. The number of days with a minimum temperature below the threshold of the 10$^{th}$ percentile (Tn10p, Fig. 10b) decreases in accordance with the consequent increase in Tmin. The relative humidity decreases (7 % i.e., 1.75 % decade$^{-1}$, Fig. 9), and this decline is completely driven by the temperature increase, as specific humidity shows no trend except a decrease in its variability during the current period (Fig.

9). Finally, there is no trend for precipitation (Fig. 9), which is linked to the fact that the thermodynamic tends to increase the precipitation while the atmospheric circulation tends to decrease the occurrence of this regime (Fig. 9).

**5.3 Blocking**

The Blocking regime is defined by a strong anticyclone over the British Isles, which blocks the inflow of maritime air and allows warm conditions to develop, especially over Western Europe. For this regime, the Paris area is isolated from the oceanic

advection, and local processes become even more influent on the climate variability. On average under this regime, only Tmin warms up significantly (1.1°C or 0.28°C decade$^{-1}$, not shown). The T2m violin plots show the same median for the two periods (Fig. 9), but a warmer mean due to the upward distribution and more hot extremes for the current period. The stretching of this side of the distribution is also observed for Tmax and Tmin (not shown). The percentage of days with a maximum temperature below the 10$^{th}$ percentile (Tx10p, Fig. 10c) decreases from approximately 4.6 % in 1979 to 1.9 % in 2017. The relative humidity

also decreases (7.2 %, i.e., 1.79 % decade$^{-1}$, Fig. 9), marked by an increase in the occurrence of events with low relative humidity. Specific humidity does not change (Fig. 9). For precipitation, there is no significant trend (Fig. 9); however, there is an increase in the frequency of rainy days and a decrease in the contribution of very wet days (Fig. 10c), which is not a



consistent result with Vautard and Yiou, 2009 at the European scale. It is the only regime in which the thermodynamical contribution to the change of precipitation is greater than the dynamical contribution (Table 2). However, these two contributions compensate each other, because the dynamical term explains 47 % of the decrease in precipitation variation whereas the thermodynamical term explains 52.1 % of the increase in precipitation variation. We observe a change of

precipitation in Blocking, which is not visible on the trends because this change is compensated by a decrease in the frequency of occurrence of the number of days in Blocking (6.2 %, ie, approximately 5.6 fewer days).

## 5.4 Atlantic Low

The Atlantic Low regime slows polar flow in favour of a southerly flow favourable to warm conditions throughout Western Europe. This regime shows the greatest changes in terms of trends (Fig. 9) from the point of view of T2m, Tmax, and Tmin,

and a strong significance of trends on temperature extremes (Fig. 10d). The T2m increases by 1.9°C (0.47°C decade$^{-1}$), the Tmax by 2.3°C (0.58°C decade$^{-1}$) and the Tmin by 1.7°C (0.42°C decade$^{-1}$). Warm and very warm anomalies increase, and cold anomalies decrease. The relative humidity decreases very strongly, by approximately 12.3 % (3.07 % decade$^{-1}$; Fig. 9), from approximately 72 % to 60.3 % in 39 years, while there is no trend for specific humidity on average. However, the median and the mean of the current boxplot are slightly lower, and the shape of the violin is strongly modified between the two periods,

with the emergence of a bimodal distribution (Fig. 9). Precipitation and extreme rainfall indices show no trends (Fig. 9 and 10d), but once again some differences between the two distributions appear, with a bimodal shape and a small increase in the occurrence of the number of days in Atlantic Low (3 %, ie, 2.7 days), accounting for 81.2 % of the precipitation change observed (Table 2).

## 5.5 The contribution of regimes to warm extremes.

Blocking and Atlantic Low are the two regimes that favour hot conditions in summer. Most heat waves over Europe occur when the Blocking or Atlantic Low regimes are installed (e.g., Cassou et al., 2005). We have seen previously that the largest trends are observed for Atlantic Low and that Blocking shows the weakest trends for temperature. If we focus on the "Summer Days" (SU; Table 1), ie, the number of days with Tmax > 25°C, the Blocking (Fig. 11 green) and Atlantic Low (Fig. 11 blue) regimes are the two regimes showing the highest frequency of SU. Figure 12 shows the evolution of the SU (number of summer

days per year) for the JJA season and for each regime. In this figure, we compute the trend for different segment sizes (minimum size of 5 years); the x-axis indicates the first year and the y-axis the final year. Red (blue) colour indicates an increasing (decreasing) trend. When considering the entire period (starting from 1979), SU increases, but this is not the case when reducing the period and starting from the middle-end of the nineties, reinforcing the idea of a temperature warming slowdown in the 2000s, although there is still no consensus on the existence of a hiatus at the global scale, with the slowdown

being the result of internal climate variability (e.g., Dai et al., 2015). The increasing trend in SU over the season (Fig. 12a) is partly due to the increase of these events during the Atlantic Low weather type (Fig. 12e), as well as in the NAO- since the end of 1990 (Fig. 12b). The Blocking regime, which is suitable to heat waves, shows a decrease of SU. This is associated with





a decrease in the frequency of Tmax ranging from 25 and 30°C, even if they are more events with Tmax above 30°C increase (Fig. 13). There is therefore an increase in episodes of very intense heat in Blocking, which is not detectable via the SU index (Fig. 12d). Similar analysis can be done for the warm Tmin (Tn90p; not shown) which is predominant in Blocking and in Atlantic Low, and which is an important factor in heat wave definition. In terms of trends, the occurrence of warm Tmin

increases in summer over the entire period, which is linked to an increase of events during the Atlantic Low from 1980 to the 2000s, followed by an increase of events during NAO- since the 2000s.

In summary, the "hot" weather regimes (Atlantic Low and Blocking) continue to contribute to extreme temperature events. However, the NAO- regime, with colder and wetter conditions compared to the first two regimes, shows strong warming trends, which leads to an increasing number of warm extremes since the 1990s, thus increasing the total probability of extreme

events in summer in the Paris area.

## 6 Discussion

On an annual scale, the climate of the Paris area has changed during the last four decades mainly due to warmer temperatures (average, maximum and minimum), with more warm extremes, fewer cold extremes and a strong decrease of the relative humidity. No significant changes are found for the specific humidity or precipitation. The rate of warming is similar to that

observed in the rest of Western Europe (Xoplaki, 2005; Van Oldenborgh et al., 2009). However, we observe a stronger increase in Tmax than in Tmin over the last 40 years, whereas Donat and Alexander, 2012 observed the opposite across different regions of the globe since the middle of the 20[th] century. In addition, they concluded that daily temperatures have become "more extreme" and that these changes are related to changes in the mean but also in the extremes; this result is also observed in our trends. One issue is to determine if the changes we found can be attributed to dynamic, thermodynamic or local anthropogenic

modifications.

### 6.1 Changes associated with large-scale dynamics

Dynamical changes are by definition related to large-scale atmospheric circulation changes. According to Vautard and Yiou, 2009, changes in atmospheric circulation are the main drivers of surface weather patterns in winter. In the Paris area over the past 40 years, we have seen very few significant trends in temperature, relative humidity and precipitation during the winter

season. Comparing the two periods, Table 3 shows that the temperature change is four times lower in winter (ΔT) than in summer. Changes in occurrence of winter regimes contribute to ¼ of the observed change versus ¾ for thermodynamic changes (Table 3). Indeed, in terms of dynamics, Yiou et al., 2018 detected significant trends in the stability of the circulation and the return period since the 1970s in winter; that is, winters tend to be similar to those already known, which increases the predictability of winter circulations. In Europe, Francis and Vavrus, 2012 and Petoukhov et al., 2013 showed that the wave

amplitude in winter is changing, particularly through a connection between the Arctic sea ice cover and the sinuosity of the jet stream which brings prolonged weather conditions enhance the probability for extreme weather as cold spell. These cold



winters may be related to the acceleration of Arctic warming associated with ice retreat (Cohen et al., 2012; Tang et al., 2013; Vihma, 2014; Walsh, 2014; Cohen et al., 2014; Zappa and Shepherd, 2017) by ice-albedo feedbacks (Screen and Simmonds, 2010). Recently, Kretschmer et al., 2018 showed that in recent decades, the stratospheric polar vortex has shifted to more frequent weak states, which may explain Eurasian cooling trends in northern winter. However, it remains controversial whether

this European winter cooling could also be related to internal atmospheric variability (Sun et al., 2016), tropical trends (Palmer, 2014), Arctic trends (Cohen et al., 2012; Tang et al., 2013; Cohen et al., 2014; Vihma, 2014; Walsh, 2014; Zappa and Shepherd, 2017), or a combination of all these variabilities. All of these processes appear to indicate that winter is marked by a stability of the circulation and that some observed trends, such as colder winters, appear to be related to modification of the atmospheric states themselves. This is why this study focuses on summer, when the changes are more significant and more related to

thermodynamical processes (+94.6 %; Table 3).

In spring, the T2m and Tmax show the strongest increase compared to the other seasons, associated with a strong decrease in RH with punctually very cold Tmax and Tmin. According to Brunner et al., 2017, this increase in temperatures in spring associated with the presence of cold extremes is also related to the position of the Blocking regime. The Blocking regime induces cold conditions in winter but warm episodes in summer. In spring, the Blocking position varies and impacts the

distribution of extreme temperatures: cold waves in early spring are induced by a Blocking position over the northeast Atlantic, while heat waves in late spring are associated with high-pressure centre over Central Europe. Cassou and Cattiaux, 2016 found a stretching of the summer period with an earlier onset of summer by ~10 days between the 1960s and the 2000s. Moreover, Boé and Habets, 2014 identified multidecadal variability with differences of river flow over France by up to 40 % in spring, which is linked with precipitation and temperature variabilities in France in spring by up to 30 % and 1°C. Part of the increase

in temperature observed in this study in the spring may therefore be associated with such multidecadal variability. Regarding precipitation in spring, the number of rainy days decreases, increasing (decreasing) dry (wet) periods, but with more extremes of rainfall. This is consistent with the multidecadal variability of the precipitation described in Boé and Habets, 2014 and Bonnet et al., 2017. These studies also suggest that these fluctuations are modulated by the Atlantic Multidecadal Variability (AMV) and that the North Pacific sea surface temperature, which exhibits variations in phase with the AMV, could also play

a role in the multidecadal variability of the main French rivers, including the Seine river, which flows in Paris. As these strong multidecadal variations can seriously impact short-term trends, it is difficult to disentangle the trends we observed that are linked to natural variability from those associated with climate change.

Another aspect is that our study area is in a transition zone in term of weather regimes. Summer regimes drive different climatic conditions at the European scale, but this distinction between regimes is not obvious when considering the Paris area, as already

shown in Dione et al., 2017. For instance, the Blocking regime is often considered as favouring heat waves, but in Paris, it is characterized by cold extremes and mean values of T2m, Tmax and Tmin closer to those of the regimes favouring colder conditions (Atlantic Ridge and NAO-). Such uncertainties are also found in precipitation, despite the use of different indices that allow the identification of the contribution of dynamics in the change of precipitation characteristics, as we did in Section 5.1. However, for more local studies, it would be interesting to carry out a sensitivity analysis on the size of the domain to be



taken into account in the calculation of weather regimes, as was done by Jézéquel et al., 2018, to select the best analogues for studying specific events in Western Europe. These results confirm that the dynamical component in climatic variability is very strong and must be taken into account, but that the thermodynamical component also plays a very important role.

**6.2 Changes associated with thermodynamic and radiative processes**

In summer, the temperature strongly interacts with the water cycle (Van Oldenborgh et al., 2009). Vautard and Yiou, 2009 even show that in summer, atmospheric circulation changes are not the main factors of surface weather trends. Over Europe, Sousa et al., 2018 analysed different forcing mechanisms associated with Blocking and Atlantic Ridge regimes, and they showed the importance of horizontal and vertical advection processes on summer temperature anomalies, especially diabatic heating processes. Although we found some changes in the occurrence of the four summer weather types, we also observed a strong evolution of the characteristics of each weather type. Table 3 shows that in summer, the average temperature change is +0.82°C between the two periods. If the thermodynamical component were the only contribution to change, this increase would have been 0.87°C; conversely, if the dynamical component, ie, the change in the occurrence, were the only contribution to the change, then we would observe a very slight decrease in temperature of approximately 0.05°C.

In summer in the Paris area, T2m, Tmax and Tmin increase strongly due to high changes in temperature extremes, while relative humidity decreases strongly with more dry anomalies. Vogel et al., 2017 show that the projected regional Tmax response in several mid-latitude terrestrial regions can be divided into (i) the global mean warming trend and (ii) an additional temperature increase, strongly influenced by soil temperature feedbacks, linked to increasingly dry soil. They also show that this feedback is mostly related to multidecadal trends in soil moisture rather than its subseasonal or interannual variability and contribute to more than 70 % of the additional warming of regional hot extremes beyond global mean warming. At the Paris scale, surface layer drying is observed from spring to autumn, as shown in Figure 14 by plotting the relationship between the seasonal surface temperature and specific humidity for each year; in this figure, colder colours are for older years and warmer colours for more recent years. In winter (Fig. 14a), there is a linear relationship between seasonal averaged T2m and q2m, meaning that if the seasonal temperature of one winter is higher, there is also more humidity and vice versa; we can even almost predict the value of seasonal humidity. It is not obvious that the more recent years have higher seasonal temperature for this season. For summer (Fig. 14c), as for winter, the temperature increase at the seasonal scale is not obvious. However, in recent years from the end of 1990s, for a similar temperature as older years, the specific humidity shows lower average values. For spring (Fig. 14b), this lack of humidity starts slightly later, from 2000, but we can also see that more recent springs present higher seasonal temperatures than older years, which amplifies the departure from the linear relationship. In Spain, Vicente-Serrano et al., 2014 observe the same trends with an increase in temperature leading to a decrease in relative humidity which is not accompanied by an increase in the surface water vapor content. They show that these trends are related to two constraints : 1) a terrestrial constraint related to a decrease of the precipitations and a decrease of soil moisture ; 2) an oceanic constraint related to a limitation in the advection of moisture from ocean surfaces.



Through the analysis of future projections, Cattiaux et al., 2015, show that the variation in diurnal temperatures increases in summer due to the decrease in surface evapotranspiration (linked to the European summer drying) and the reduction in cloud cover. This variation in diurnal temperatures is already observed in the Paris area with an increase in Tmax above Tmin observed in spring and summer. In autumn (Fig. 14d), there are also lower humidity values for similar temperatures, but the

signal is weaker than for summer. However, there is a striking increase in seasonal temperature for this season, associated with a nearly linear increase of humidity, unlike MAM. For the recent period, the warmer autumn seasonal averages associated with higher specific humidities are notably due to warmer and moister November months. At the seasonal time scale on the SIRTA supersite near Paris, Bastin et al., 2018 show that temperature variability is mainly controlled by surface fluxes.

At the Paris scale, in summer, the total rain amount increases but not significantly, and there is no change in specific humidity.

The link between increasing temperatures (seen previously) and increasing precipitation has been highlighted by Rasmussen et al., 2017, who show, using a climatic simulation at convective-permitting resolution, the change in convective population in a warmer future climate, induced by both the increase of the CAPE (Convective Available Potential Energy) but also of the CIN (Convective INhibition). Convection becomes more difficult to trigger, but once triggered, the energy available for convection is increased, favouring heavier precipitation. This means that weak to moderate convection will decrease and strong

convection will increase in frequency in a future climate. This result therefore leads to a modification of the spatial and temporal occurrence of the precipitations. This may explain the bimodal structure displayed by the violin plot of humidity and precipitation in Fig. 9 for the Atlantic Low regime. The link between soil moisture and precipitation remains poorly understood. Indeed, Boé, 2013 shows that in summer in France, previous soil moisture conditions could have a limited impact on precipitation through a modulation of large-scale circulation and the absolute effect of soil moisture on evapotranspiration is

much larger than its effect on precipitation. Additionally, Vogel et al., 2017 show that changes in precipitation can also influence temperature and soil moisture variations.

Within the summer season, we observe very significant changes over 40 years independent of large-scale circulation, thus raising questions about the role played by local surface-atmosphere feedbacks in the context of warming. Temperatures in regimes favouring "cold conditions" warm up very clearly and even contribute for some years to very warm temperatures.

Regimes favouring "hot conditions" continue to contribute very significantly to extreme heat events, such as heat waves. In particular, the Atlantic Low regime shows a very strong increase in the temperatures and a very strong decrease in the relative humidity, whereas precipitation and specific humidity show no trend but a change of their distribution. In a recent study, Bastin et al., 2019 analysed the spatial variability over Europe of the temperature thresholds over which the relative humidity starts to decrease using an integrated water vapour dataset from GPS stations. They suspect that the spatial variability of this

threshold is strongly linked to local processes that drive moisture sources, in particular surface-atmosphere interactions and coastal/orographic circulations.

As shown by Zampieri et al., 2009 and Cattiaux et al., 2012, recent and future warming in Europe are incompatible with changes in atmospheric circulation alone, and surface-atmosphere processes are the mainly responsible for increasing



temperature variability, especially summer temperatures. Furthermore, uncertainties in regional temperature projections can be linked to this long-term soil moisture-temperature feedback (Vogel et al., 2017).

### 6.3 Changes associated with local anthropogenic effects

Finally, some of the changes detected can be attributed to anthropogenic influence on land use, such as urbanization and
irrigation. Changing a vegetated area to pavement strongly modifies the surface processes, with more run-off, less evapotranspiration, and more heat. The enlargement of suburban areas affects the urban heat island processes. This is not the purpose of the paper, but it is an indispensable aspect to discuss in observed changes. Daniel, 2017 compares different representations of urban areas within an atmospheric model with an explicit representation of the urban areas and concludes that cities can influence their environment on a regional scale. Thus, the largest French cities induce a warming trend of the
temperature near the surface. This warming can reach up to 1.5°C in summer Tmin in Paris. Thus, according to Wilcox et al., 2018, anthropogenic forcing may have slightly increased the risk of dry summers and greatly increased the risk of hot summers.

### 7 Conclusion

This study characterizes the main changes in trends and extremes of temperature, humidity and precipitation at the local scale in the Paris area, which is favoured by an urban heat island. The analysis was carried out annually and seasonally, including
the effect of large-scale circulations in summer. The comparison of the observations with the ERA-I reanalysis shows that it strongly underestimates the temperatures (especially in summer) and overestimates the relative humidity. The local trends are not adequately characterized by ERA-I, especially for the climate extreme indices. This analysis thus confirms the importance of direct observation when dealing with local scale. This study uses observation data from 5 stations in the Paris area since 1979 to characterize observed changes in temperatures, relative humidity and precipitation at different time scales. Although
some trends are similar to the ones found at regional (Europe) or global scales, there are specific local patterns:

- Tmax increases more strongly than Tmin at annual, seasonal (except SON) and summer scales.
- There are few significant trends in winter, unlike in summer.
- Summer temperatures increase due to a strong thermodynamical contribution.
- In summer, the temperatures increase for the cooler weather regimes, especially NAO-, contributing to high
temperatures. During this time, the hottest weather regimes keep warming even more. This is due to the advection of warming air masses from the ocean and a probable intensification of temperature in the air.
- The relative humidity decreases considerably, especially in spring and summer. This is particularly true for the Atlantic Low weather regime in summer.
- The specific humidity shows little or no trends, although it was expected to increase associated with warming, and
the proximity to the English Channel.



- Rainfall has a high variability from one year to the next, but the trend, even if not truly significant, appears to be decreasing (except in summer). There appears to be a change in the precipitation regime with a less rainy winter, a generally drier spring with more intense rainfall and a wetter summer, mainly due to a change in occurrence in summer weather regimes.

It is important to understand the physical processes behind these changes at the local scale and especially during the summer season, as they are likely to intensify or become inhibited with the current climate change. Some of these processes have been discussed in the previous section; however, there are several feedbacks that are still poorly understood in the context of global warming, particularly in such an urbanised area. As this very recent study by Schwingshackl et al., 2018 shows, it is crucial to take into account local and regional processes to properly assess inter-annual variability in temperature and future trends in temperature.

One of the perspectives of this study is to understand these current changes using the rather complete set of atmospheric observations from the supersite of SIRTA (Chiriaco et al., 2018), which collects more than 50 meteorological and atmospheric parameters at hourly time steps since 2003 over the full boundary layer. The strong correlation between the stations as well as the average of the stations encourages us to use this dataset.

**Author contributions**

JR carried out the data analysis and prepared all the figures. JR, MC, SB and FH contributed to the data analysis and interpretation of results. JR wrote the manuscript with contributions from all co-authors.

**Competing interests**

The authors declare that they have no conflict of interest.

**Acknowledgements**

The study was supported by the LABEX L-IPSL, funded by the French Agence Nationale de la Recherche, under the programme « Investissements d'Avenir » (grant ANR-10-LABX-18-01). We would like to acknowledge Météo France for the provision of observation measurements and the SAFRAN analysis. P. Yiou is supported by the ERC grant 338965-A2C2. To process the data, this study benefited from the IPSL mesocenter ESPRI facility which is supported by CNRS, UPMC, Labex L-IPSL, CNES and Ecole Polytechnique.



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





**Figure 1: Study area in the Paris region (France). Yellow: the Météo France observation stations (OBS). Cyan: the SIRTA super site. The green area represents the ERA-I coverage (4 pixels) and the orange area reflects the SAFRAN coverage (36 detailed pixels not shown).**





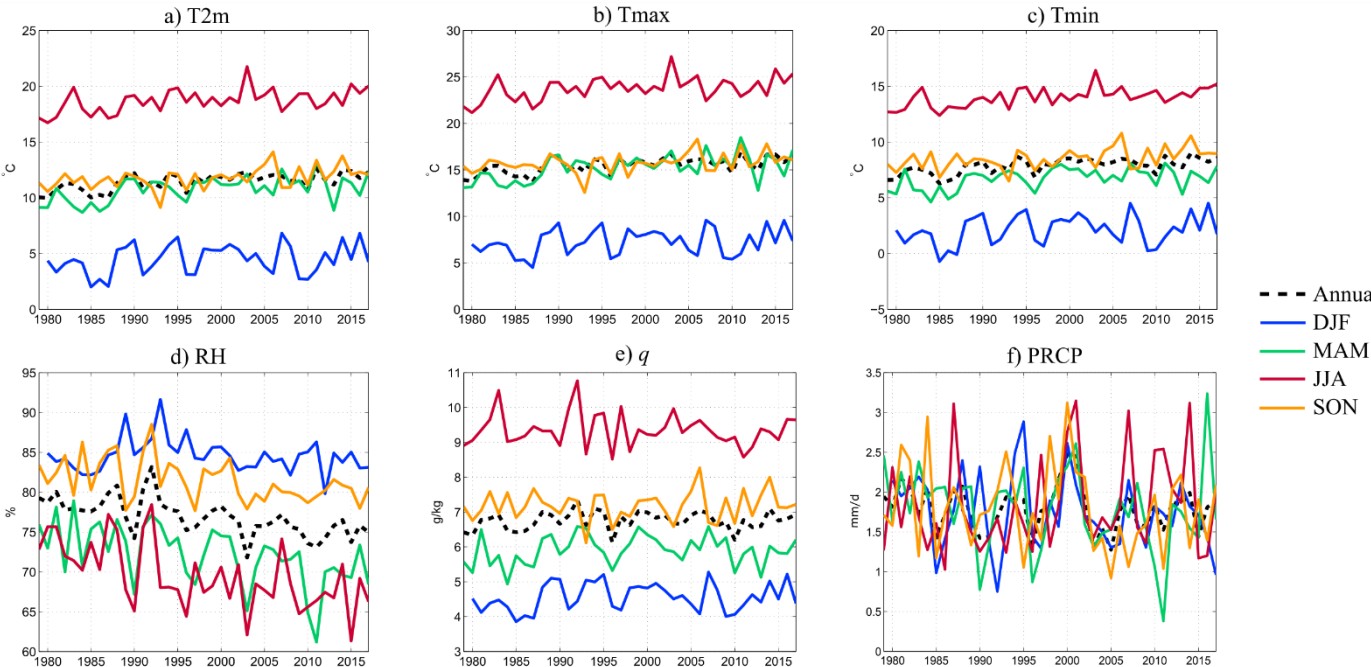

**Figure 2: Annual averages (dashed black line) and seasonal averages (coloured lines) of T2m (°C), b) Tmax (°C), c) Tmin (°C), d) RH (%), e) $q$ (g.kg$^{-1}$) and f) daily PRCP average (mm.d$^{-1}$). DJF is in blue, MAM in green, JJA in red, and SON in orange.**

| Index | Name | Definition | Units |
|---|---|---|---|
| SU | Summer Days | Annual count of days when Tmax> 25 ° C | days |
| ID | Icing Days | Annual count of days when Tmax < 0°C | days |
| Tx90p | Warm Days | Percentage of days when Tmax > 90$^{th}$ percentile | % |
| Tx10p | Cool Days | Percentage of days when Tmax < 10$^{th}$ percentile | % |
| Tn90p | Warm Nights | Percentage of days when Tmin > 90$^{th}$ percentile | % |
| Tn10p | Cool Nights | Percentage of days when Tmin < 10$^{th}$ percentile | % |
| TR | Tropical Nights | Annual count of days when Tmin > 20°C | days |
| FD | Frost Days | Annual count of days when Tmin < 0°C | days |
| %Rainy | Annual rainy days | Percentage of days when RR>0.2 mm | % |
| R90pTOT | Very wet days | Annual total PRCP when RR > 90$^{th}$ percentile | mm |
| PRCPTOT | Annual total wet-day precipitation | Annual total PRCP in wet days (RR > 0.2 mm) | mm |



| SDII | Simple daily intensity index | Annual total precipitation divided by the number of wet days | mm day$^{-1}$ |
| CWD | Consecutive wet days | Maximum number of consecutive days with RR ≥ 0,2 mm | days |
| CDD | Consecutive dry days | Maximum number of consecutive days with RR < 0,2 mm | days |

**Table 1: Climate indices (temperature in the 8 first lines and precipitation in the 6 last lines) based on Climdex indices. In this study, the threshold between a dry day and a rainy day (RR) is set at 0.2 mm day$^{-1}$, unlike ETCCDI, which uses a threshold of 1 mm day$^{-1}$.**

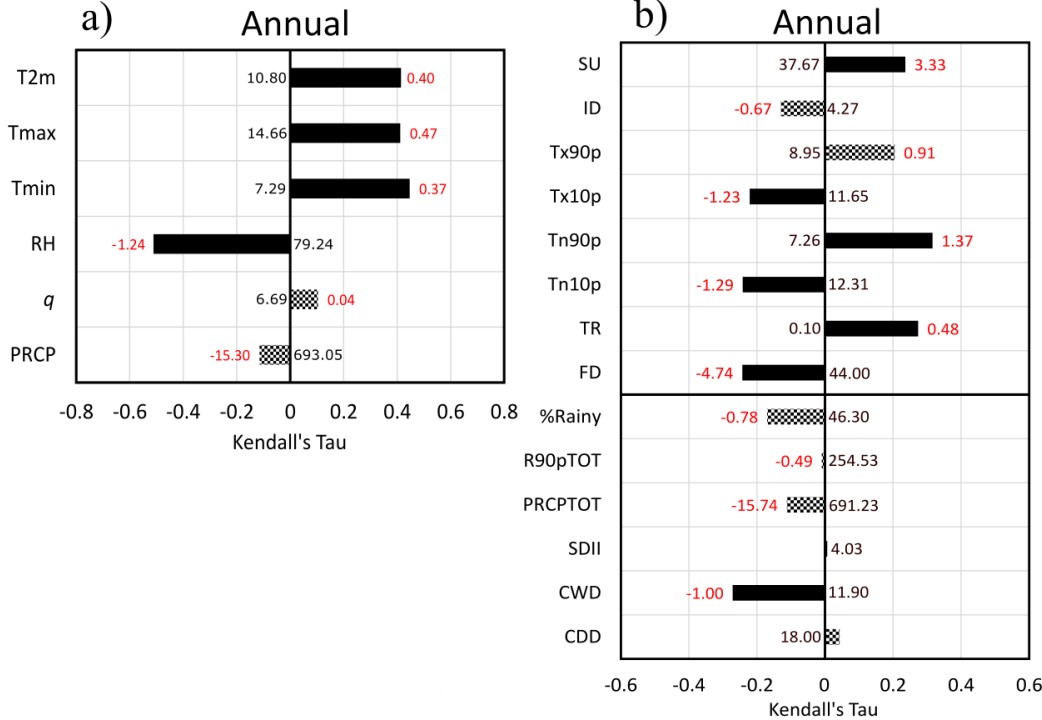

5  **Figure 3: Mann-Kendall annual trends in observational data for (a) T2m, Tmax, Tmin, RH, *q* and PRCP, and for (b) climate indices from Tmax, Tmin and precipitation. On the abscisse, Kendall's Tau represents the rank correlation coefficient between the variable and time. The red value represents the Sen slope, i.e., the median slope in units per decade, and the black value represents the average original value in 1979 (in units). A solid bar indicates a significant trend for a confidence interval of *p* = 0.05, and a mosaic bar indicates a non-significant trend.**




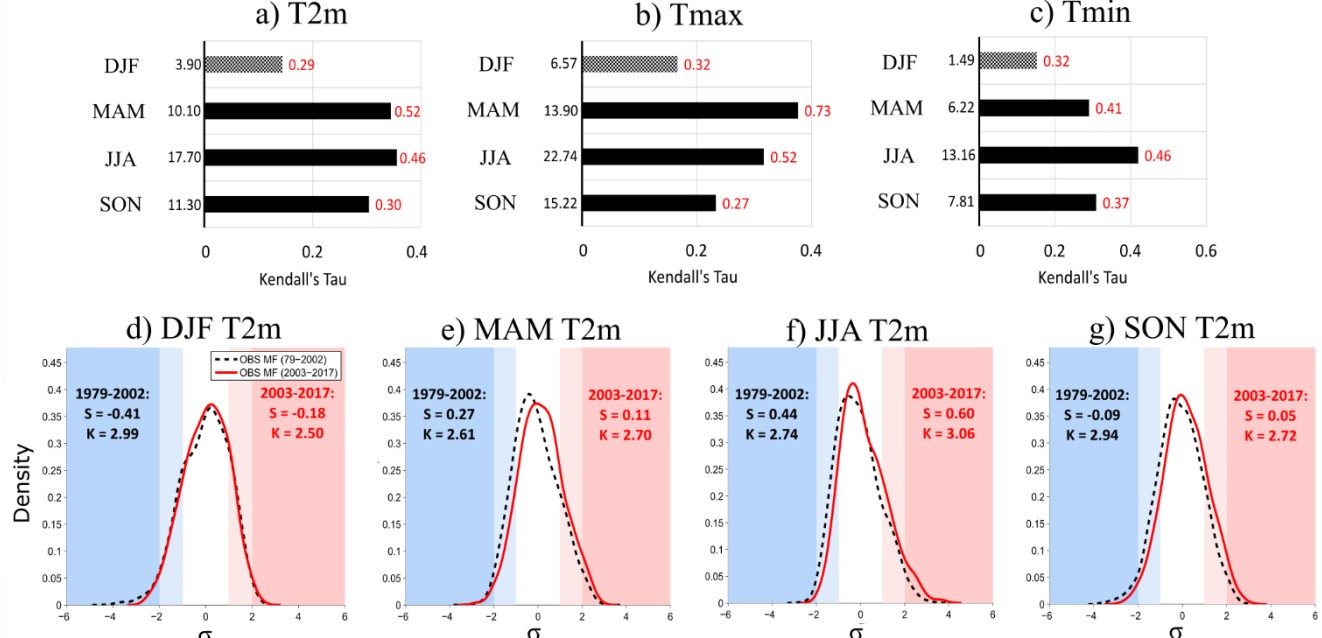

**Figure 4: Top, Mann-Kendall seasonal trends in observational data for (a) T2m, (b) Tmax and (c) Tmin. Characteristics of the figure are the same as for Fig. 3. Bottom, seasonal PDF of the daily anomalies of T2m, normalized over the period 1979-2017, for d) DJF, e) MAM, f) JJA and g) SON. Dashed black: the past period from 1979 to 2002; red line: the current period from 2003 to 2017. For each period is calculated the symmetry coefficient (S) and the shape coefficient (K). The white part of the figure corresponds to [-1 <σ <+1], light colours to [-2 <σ <-1; 1 <σ <2], and dark colours to [σ <-2; σ> 2].**




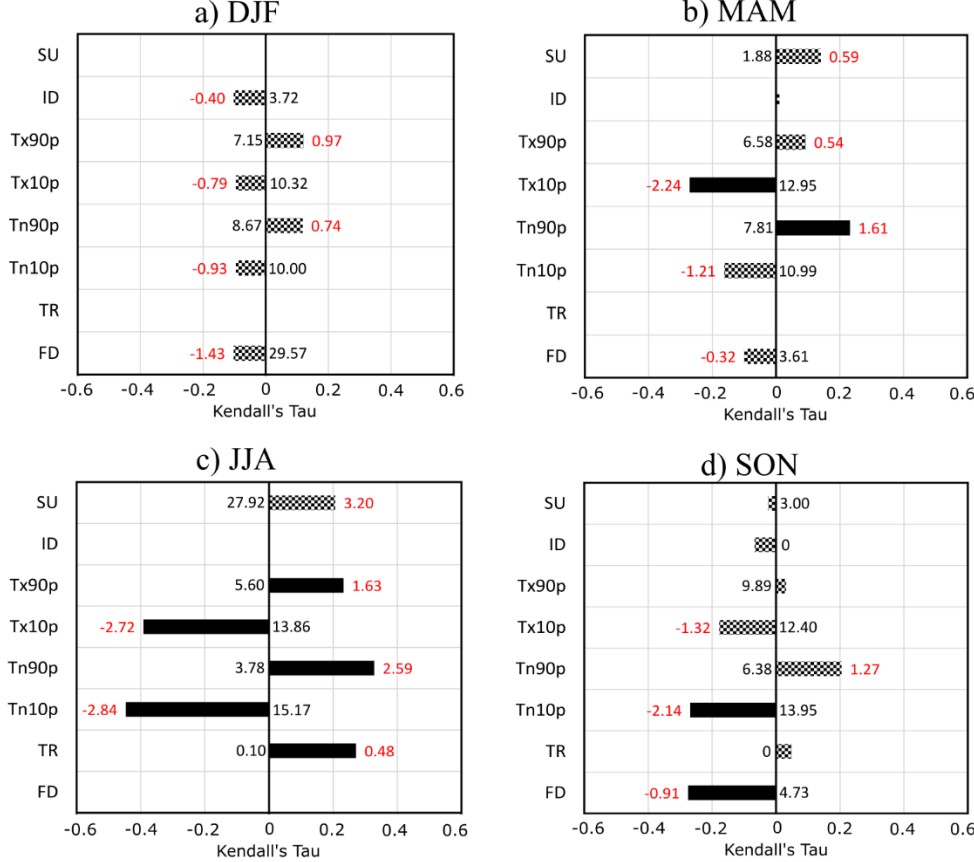

**Figure 5: Mann-Kendall seasonal trends of temperature climate indices calculated from Météo France observations stations for the four seasons (a) DJF, (b) MAM, (c) JJA, (d) SON. See Table 1 for temperature climate indices. On the abscisse, Kendall's Tau represents the rank correlation coefficient between the variable and time. The red value represents the Sen slope, i.e., the median slope in units per decade, and the black value represents the average original value in 1979 (in unit). A solid bar indicates a significant trend for a confidence interval of $p = 0.05$, and a mosaic bar indicates a non-significant trend.**





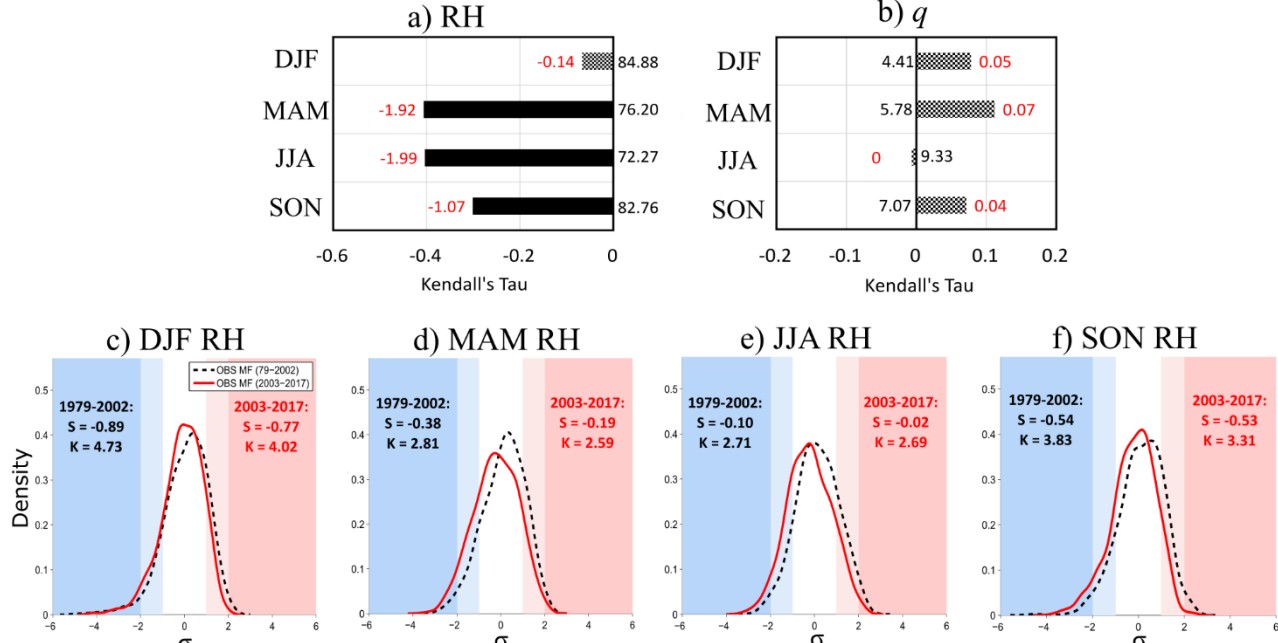

**Figure 6: Same as Fig. 4 for RH (a, c, d, e, f) and q (b).**

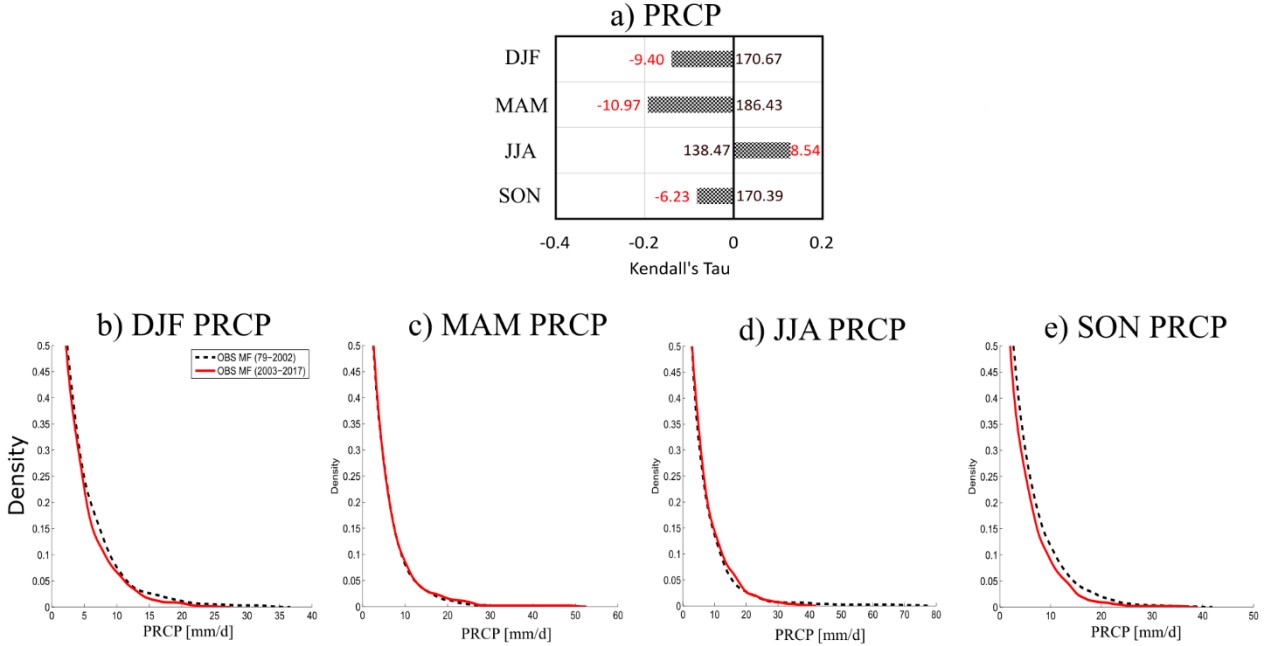

**Figure 7: Top, same as Fig. 4a but for PRCP. Bottom, seasonal PDF of daily intensities of rainy days only (> 0.2 mm day⁻¹) for b) DJF, c) MAM, d) JJA and e) SON. Dashed black: the past period from 1979 to 2002; red line: the current period from 2003 to 2017.**



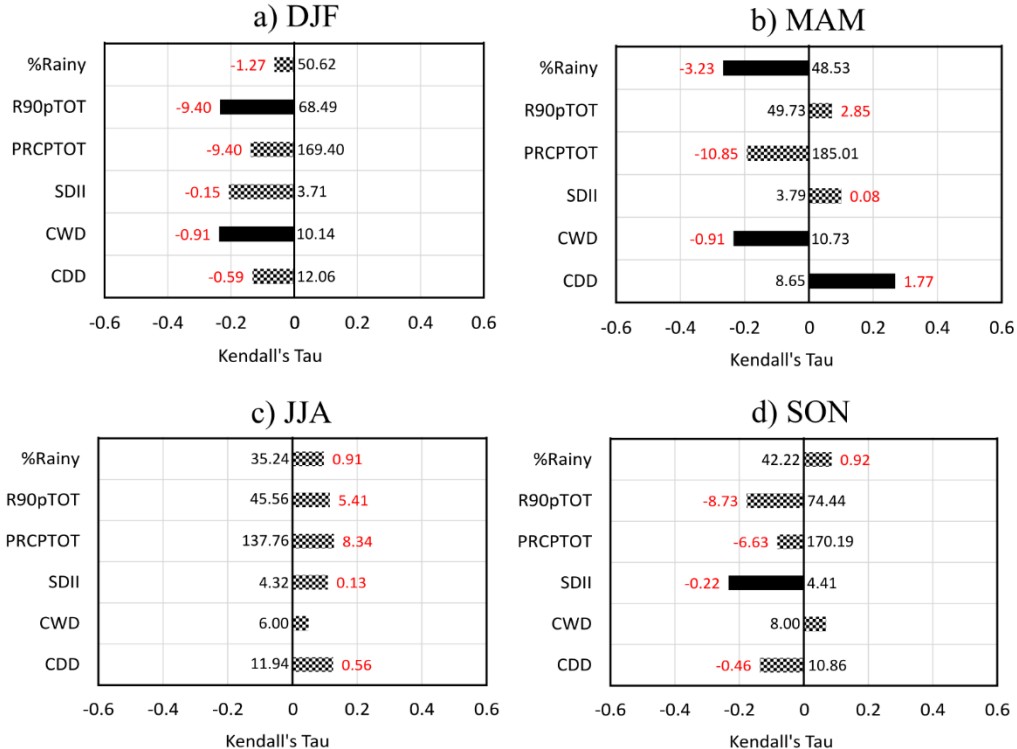

**Figure 8: Same as Fig. 5, but for precipitation climate indices.**

| | Dynamical contribution [mm (%)] | Thermodynamical contribution [mm (%)] | $\Delta PRCP$ [mm] |
|---|---|---|---|
| SUMMER (JJA) | 5.3 (32.4) | 11.1 (67.6) | 16.4 |
| | Dynamical contribution [mm (%)] | Thermodynamical contribution [mm (%)] | $\Delta PRCP_i$ [mm] |
| NAO- | 20.4 (87.7) | -2.9 (-12.3) | 17.5 |
| Atlantic Ridge | -10.3 (-74.2) | 3.6 (25.8) | -6.7 |
| Blocking | -8.7 (-47.9) | 9.5 (52.1) | 0.8 |
| Atlantic Low | 3.9 (81.2) | 0.9 (18.8) | 4.8 |

**Table 2: Dynamical and thermodynamical contributions of the precipitation change ($\Delta PRCP$) in mm for summer (JJA) and for the four weather regimes in summer. Values in parenthesis give the ratio (in %) between the change components and the total change.**





**Figure 9: Violin plot of daily T2m (first line), RH (second line), *q* (third line) and PRCP (fourth line) for the four summer weather regimes between the periods 1979-2002 and 2003-2017 (one regime, one column). The black bar represents the mean, and the red bar the median. Boxed numbers represent trends in unit decade$^{-1}$ over the period 1979-2017. The asterisk represents a significant trend for a confidence interval of *p* = 0.05.**





**Figure 10: Mann-Kendall trends in observational data for climate indices for the four summer weather regimes a) NAO-, (b) Atlantic Ridge, (c) Blocking and (d) Atlantic Low. Figure characteristics are the same as for Fig. 7.**



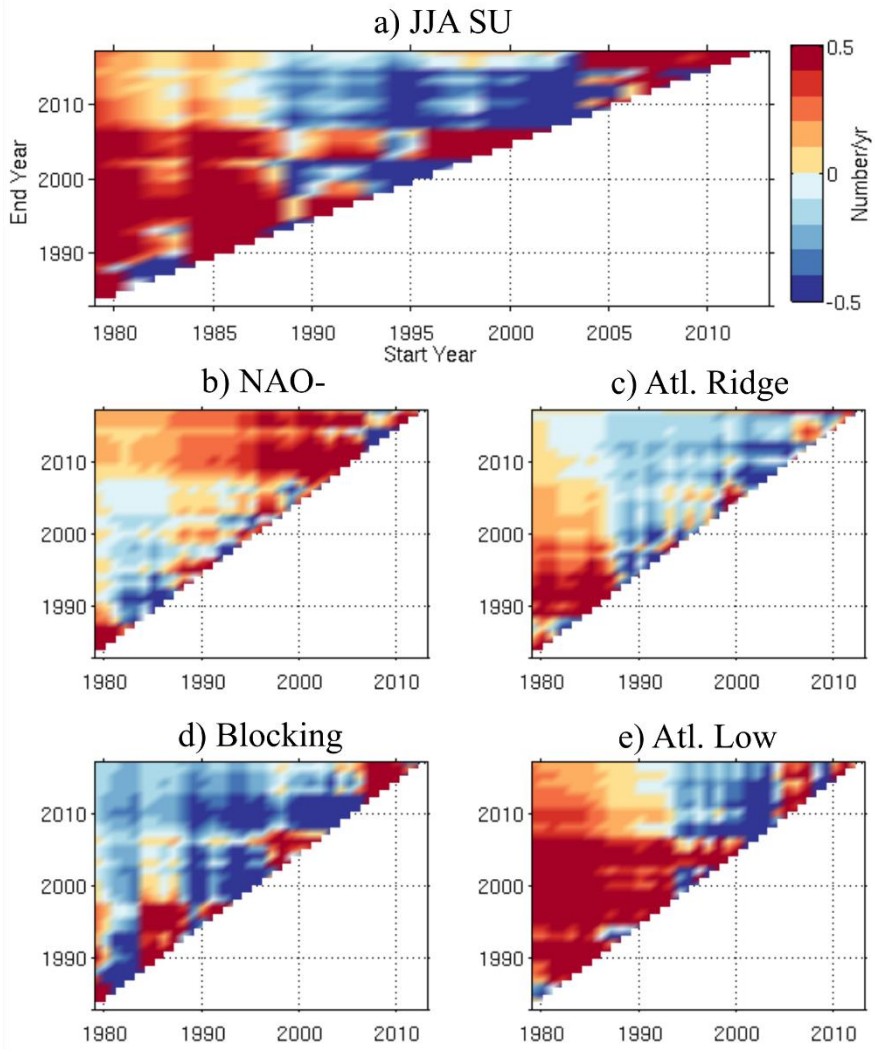

**Figure 11: "Summer Days" frequency (Tmax >25°C) in number of days for the JJA season (black boxplot) and for each summer weather regime calculated over the period 1979-2017.**

**Figure 12: Linear trends of SU (in number of day yr⁻¹) as a function of length of segment (y-axis represents the end year of the segment, and x-axis represents the starting year of the segment) for a) Summer, b) NAO-, c) Atlantic Ridge, d) Blocking and e) Atlantic Low. The minimum segment size is 5 years, and the trend is calculated by linear regression.**



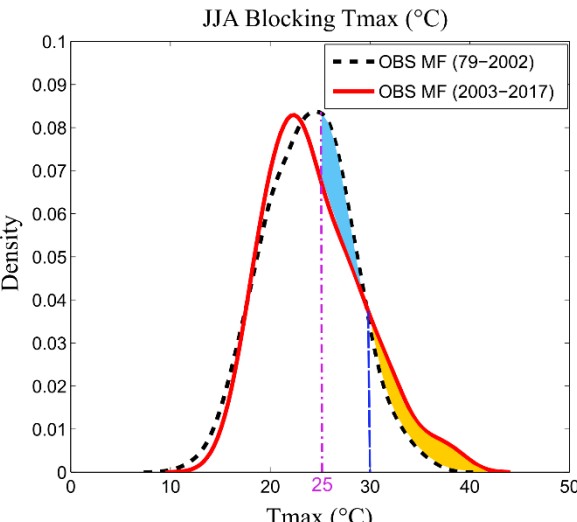

**Figure 13: PDF of the Tmax for the Blocking regime. The dotted black line represents the past period from 1979 to 2002, and the solid red line reflects the current period from 2003 to 2017. The purple vertical segment represents the threshold of the SU (Summer Days), i.e., 25°C. The blue vertical segment represents the temperature at which a frequency inversion occurs between the past period, with a higher frequency of temperatures between 25 and 30°C (blue coloured zone), and the current period, with a higher frequency of temperatures above 30°C (zone coloured orange).**

| | Dynamical contribution [°C (%)] | Thermodynamical contribution [°C (%)] | $\Delta T$ [°C] |
|---|---|---|---|
| WINTER (DJF) | 0.06 (*27.3*) | 0.17 (*72.7*) | 0.23 |
| SUMMER (JJA) | -0.05 (*-5.4*) | 0.87 (*94.6*) | 0.82 |

**Table 3: Dynamical and thermodynamical contributions of the temperature change ($\Delta T$) in °C in winter (DJF) and in summer (JJA). Values in parenthesis give the ratio (in %) between the change components and the total change.**





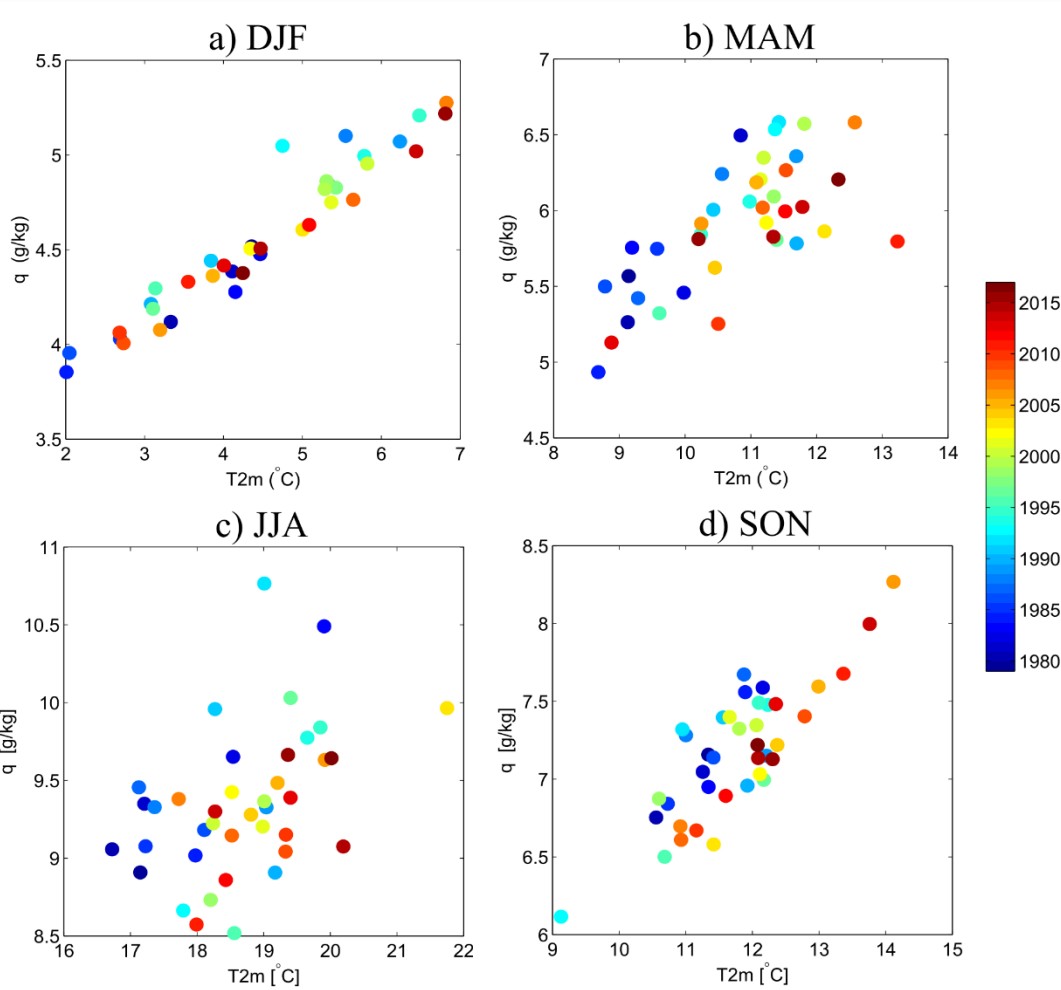

**Figure 14: T2m (°c) – *q* (g kg⁻¹) seasonal relationship in the Paris area from observations. Each point represents the seasonal average of one year.**



## Appendix A: Comparison of the local observation with the ERA-Interim reanalyses and SAFRAN analysis

### ERA-Interim

The ERA-Interim reanalysis (ERA-I) developed by the European Centre for Medium-Range Weather Forecasts (ECMWF) is a global atmospheric reanalysis available from 1979 to today, every 6 hours and at a spatial resolution

of 0.75°x0.75° (Dee et al., 2011; Dee and Uppala, 2009; Simmons et al., 2014). The ERA-I dataset contains both analyses and forecasts. Unlike T2m, which contain analyses four times per day (00 h, 06 h, 12 h and 18 h), Tmax and Tmin series under ERA-I are built from daily forecasts. There are five time values per day: 0, 6, 12, 18 and 24 h, corresponding to the 5 forecasting steps (12, 18, 24, 30, and 36 h) starting at the reference time 12 h of the day before; thus, the daily value Tmax and Tmin of ERAI are selected by selecting the maximum or minimum daily

values from the 5 values available on the corresponding day. The configuration of the ERA-I grid imposes a grid point in the near centre of our study area, involving the presence of observation stations on four different ERA-I pixels (green square, Fig. 1). We performed a sensitivity analysis to compare each pixel to the average of the four pixels. The result shows that the differences between each pixel, as well as the average of the four pixels, is very weak for all considered variables. We only observed a slightly different variability for the northwestern pixel at the

seasonal scale, with a slightly warmer and drier pixel for T2m in winter and a little colder and slightly wetter pixel for T2m and Tmax in the summer. This pixel is located closer to the English Channel (only 55 km), so it is more subject to oceanic conditions with milder winters and cooler summers. We chose to average the four pixels in order to obtain a spatial coverage including all the observation stations. The data of T2m, Tmax, Tmin and RH are collected for the four ERA-I pixels (green square, Fig. 1) and then averaged to obtain a daily spatial average.

### SAFRAN

For precipitation, we use a meteorological analysis system named SAFRAN (*Système d'Analyse Fournissant des Renseignements Adaptés à la Nivologie*) (Durand et al., 1993) developed by the *Centre National de Recherches Météorologiques et le Centre d'études de la Neige* (CNRM/CEN). The main characteristic of SAFRAN is its treatment of a limited area divided into non-regular and climatologically homogeneous areas. As input, SAFRAN

uses vertical profiles derived from the meteorological model as well as numerous sources of observations. The data are analysed by altitude range (300 m steps) via optimal interpolation (6 h time steps, and 24 h time steps for precipitation). The analyses are then interpolated at the hourly time step; then, a spatial interpolation is performed to project the data on a regular grid. In output, the SAFRAN meteorological analysis system has a spatial resolution





of 8 km x 8 km and an hourly temporal resolution. These data are available from 1958 to 2016. This study collects and averages 36 pixels, whose spatial coverage represents the "small Parisian crown" (orange in Fig. 1).

**Statistical comparisons**

The daily average of the five Météo France observation stations for T2m, Tmax, Tmin, RH and $q$ are compared to the daily average of the 4 pixels of ERA-I grid, which encompasses the MF stations (in green in Fig. 1). For precipitation, we compare the daily average of the four MF stations (without Trappes) with the daily average of the SAFRAN grid (in orange in Fig. 1). The statistical comparison uses the correlation coefficient, the bias and the standard deviation.

On an annual scale (Fig. A1 a-c, Appendix A), all variables except precipitation show a very good correlation coefficient between observations and analysis (Fig. A1a). ERA-I underestimates T2m, especially Tmax with -1°C (Fig. A1b), and overestimates RH (about +4 %). The standard deviation from the diagonal is very small for temperatures and specific humidity but more significant for relative humidity (Fig. A1c). For precipitation, SAFRAN bias is very low (Fig. A1b), but probably due to compensatory errors, since the correlation coefficient is not very high and associated with a significant standard deviation (Fig. A1c).

At the seasonal scale (Fig. A2 d-f), the correlation coefficient for temperatures and humidities is very good in all seasons, but not for precipitation (Fig. A1d). This is certainly a signature of the very high variability of precipitation. The annual underestimate of temperature by ERA-I is the result of an underestimate of temperatures for all seasons. It is, however, more significant in summer, with -1.4°C for Tmax and -1°C for T2m (Fig. A1e). This strong underestimation of ERA-I is also marked in spring and for Tmax in autumn (Fig. A1e). Several reasons explain this underestimation of ERA-I on the temperatures. First, the coverage taken into account of the ERA-I grid is greater than the "Parisian crown". Second, for T2m, the daily temperatures are averaged over the analyses performed every 6 hours. Third, Tmax and Tmin are not analyses but daily forecasts. ERA-I overestimates relative humidity for all seasons, especially in MAM and JJA with values near 4 % (Fig. A1e). Moreover, the standard deviation from the diagonal is very strong (Fig. A1f). These are months with humidity coming from surface evaporation playing an important role in the total relative humidity amount. This overestimate by ERA-I suggests stronger latent heat flux in ERA-I than in observations. For rainfall, the SAFRAN bias is relatively low at all seasons (Fig. A1e), but summer has less correlation (Fig. A1d) and more scattering than other seasons (Fig. A1f). JJA corresponds to a period when precipitations are mostly convective, and more locally and suddenly impact the local measurements. The statistical evaluation at the daily time scale is thus very challenging. The statistical





analysis of SAFRAN performed from the monthly accumulation gives better results, confirming that SAFRAN is an analysis module that represents the precipitation at the local scale rather well.

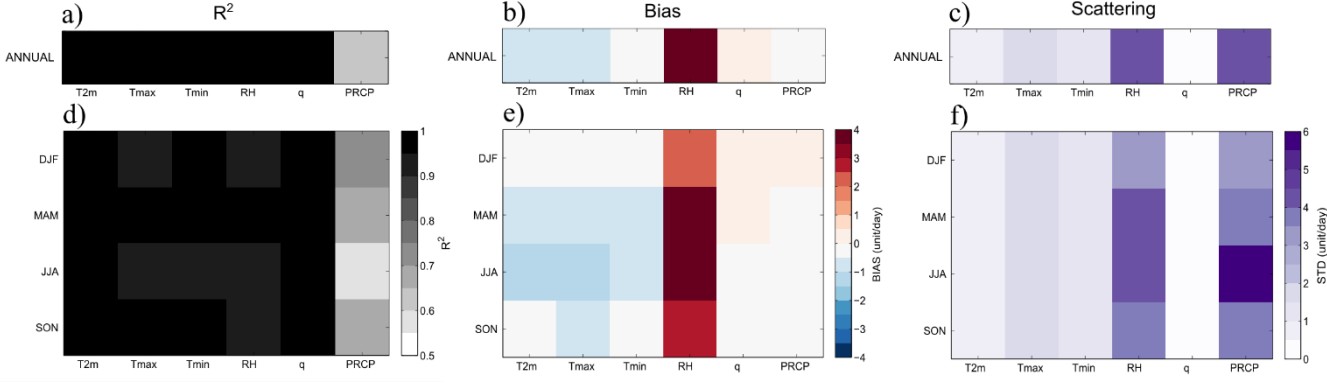

**Figure A1: Statistical comparison between daily ERA-I and daily MF observation for T2m, Tmax, Tmin, RH, and q (one column one variable in each sub-figure); and between daily SAFRAN and daily MF observation for PRCP (last column in each figure). a-c) Annual statistical comparison and d-f) seasonal statistical comparison. a) and d) For $R^2$, the correlation coefficient. b) and e) For the bias (in units day$^{-1}$). c) and f) For scattering, the standard deviation from the diagonal.**



## Appendix B: North Atlantic Weather Regimes in Summer

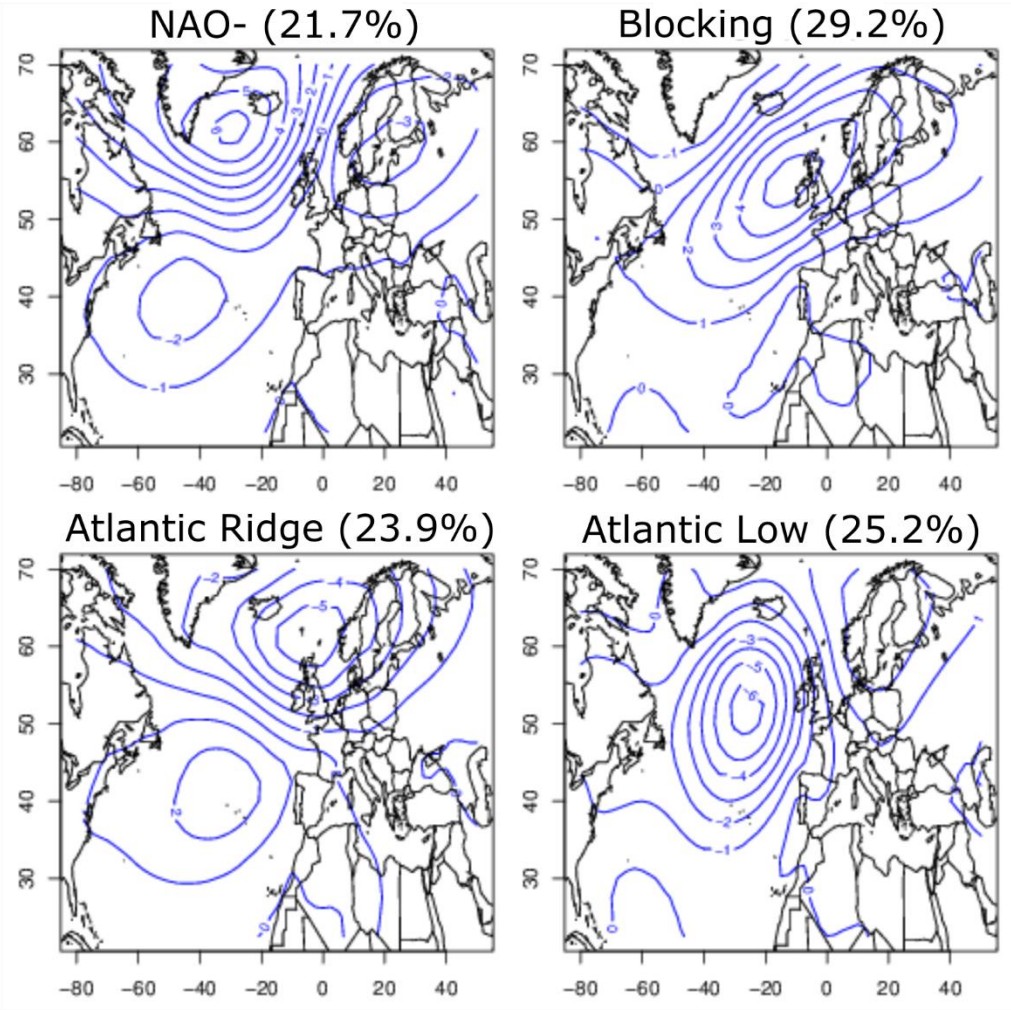

**Figure B1: Summer North Atlantic weather regimes computed on the sea level pressure from National Centres for Environmental Prediction reanalysis based on reference periods from 1970 to 2010. The weather regimes were determined on seasonal anomalies of SLP. The isolines show SLP anomalies in hPa for NAO-, Atlantic Ridge, Blocking and Atlantic Low. The average frequencies of the regimes over the 1979-2017 period are indicated by percent signs. Figure from https://a2c2.lsce.ipsl.fr/index.php/deliverables**