# Peer review of "Recent trends in climate variability at the local scale using 40 years of observations: the case of the Paris region of France"

_Atmospheric Chemistry and Physics, 2019_

## Referee Comment (RC1) · Anonymous Referee #1 · 10 May 2019

Review of: Recent trends in climate variability at the local scale using 40 years of observations: the case of the Paris region France.

Summary: The analysis of variability of local scale uses temperature, moisture, and precipitation to evaluate 40 years of observations to attempt to identify thermodynamic versus dynamical changes in extreme events. The focus is on the summer season, with companion analysis of the other three seasons.

The goal of this manuscript is to determine the differences (if any) between thermodynamic and dynamic influences on extremes. I believe this is a good question and worth exploring this topic. Dynamical vs thermodynamical constraints are important

considerations for the analysis of what drives extremes, especially within the context of climate change.

However, the authors do not use the correct definition of thermodynamics in their analysis. Temperature and humidity are analyzed independently, when there is well established literature demonstrating these variables are co-dependent. Temperature and humidity covary together, and non-linearly in extreme regimes. These can be readily calculated from Reynolds averaging (see eq. 13 from Buzan et al. 2015 for an example; shown below). From an analysis of the methods, I cannot determine if this was taken into account. The temperature and humidity plots are shown in isolation, and when they are mapped together (Fig. 14), they are based upon seasonal averages.

From the literature review, every major manuscript on temperature-moisture covariance from the past decade is missed (list of manuscripts below). To determine thermodynamically driven events, a pure thermodynamical variable should and a computed. Wet bulb temperatures/equivalent potential temperatures are linked to atmospheric convection, which demonstrated in theory (see Williams et al., 2009; figure below) and observation (Williams et al., 2017; example figure below). Wet bulb temperature maximums are directly tied to convection limits of the atmosphere. I recommend using Wet bulb temperature as they lead to clear thermodynamic events. However, there are other moist-temperature variables that could be utilized in the context of extremes such as evaporative cooling efficiencies (swamp cooler temperatures), heat stress metrics (e.g. Wet Bulb Globe Temperature which is temperature-radiation-moisture covariance). Overall, analyzing temperature and moisture independently fundamentally omits the total thermodynamical regime. I believe there are many more figures than are necessary for the analysis resulting from treating these state variables as independent.

Overall, I believe there are fundamental missing characteristics that are necessary to show thermodynamic vs dynamically driven changes in variability for the past 40 years. There is a lot of potential in this manuscript, and I am interested in evaluating a future manuscript that handles the full thermodynamics. Many of the same techniques

evaluating maximums and correlating events would apply when using a pure moist thermodynamic variable.

Ex. 1) The observation stations (p. 4, line 8) shows warm-drier conditions Montsouris vs cooler-moist Trappes. The area between these stations is relatively small, and a metric such as wet bulb temperature or virtual temperature, I suspect, would show almost no difference between these stations. This is due to the moist-temperature covariance is fundamentally connected to equivalent potential temperatures, which are tied to the entropy state of the region (in other words, there are not significant differences in the total energy between the two stations).

2) Using wet bulb temperatures would remove the RH and q plots, and produce precipitation correlations between thermodynamics and dynamical driven processes. For example, figure 9 would consist of Tw and precipitation. Maximum wet bulb temperatures show clear pdf properties and have a clear non-gaussian shape (see Sherwood and Huber, 2010 figure 1E; posted below), thus the changes in Figure 13 should show clearer separations between the two time periods.

Supplemental: calculating moist-temperature values can be difficult. Choice of algorithms, especially in the context of extremes, can impact the results. I have attached code from the HumanIndexMod from the manuscript Buzan et al., 2015. The Davies-Jones wet bulb code is computationally fast and accurate to extreme conditions. The code is fortran designed to work with NCL fortran wrapper. T,P,Q are required; winds are optional.

References:

D Bolton. The computation of equivalent potential temperature. Monthly  Weather Review, 108(7):1046–1053, 1980.   J R Buzan, K Oleson, and M Huber. Implementation and comparison of a suite of heat stress metrics within the Community Land Model version 4.5. Geosci. Model Dev., 8(2):151–170, 2015.   Michael P Byrne and Paul A O'gorman. Link between land-ocean warming contrast and surface relative

humidities in simulations with coupled climate models. Geophysical Research Letters, 40(19):5223–5227, October 2013.

Robert Davies-Jones. On Formulas for Equivalent Potential Temperature. Monthly Weather Review, 137(9):3137–3148, September 2009.   Robert Davies-Jones. An Efficient and Accurate Method for Computing the Wet-Bulb Temperature along Pseudoadiabats. Monthly Weather Review, 136(7):2764–2785, July 2008.   Noah Diffenbaugh, Jeremy Pal, Filippo Giorgi, and Xuejie Gao. Heat stress intensification in the Mediterranean climate change hotspot. Geophysical Re- search Letters, 34(11):L11706, 2007.   E M Fischer, K W Oleson, and D M Lawrence. Contrasting urban and rural heat stress responses to climate change. Geophysical Research Letters, 39(3):1– 8, February 2012.   Carlos D Hoyos and Peter J Webster. Evolution and modulation of tropical heating from the last glacial maximum through the twenty-first century. Climate Dynamics, 38(7-8):1501–1519, November 2011.

L S Kalkstein and J S Greene. An evaluation of climate/mortality relationships in large U.S. cities and the possible impacts of a climate change. Environmental Health Perspectives, 105(1):84, 1997.   Robert L Korty, Kerry A Emanuel, Matthew Huber, and Ryan A Zamora. Tropical Cyclones Downscaled from Simulations with Very High Carbon Dioxide Levels. J. Climate, 30(2):649–667, January 2017.

Rahul Kumar, Vimal Mishra, Jonathan Buzan, Rohini Kumar, Drew Shindell, and Matthew Huber. Dominant control of agriculture and irrigation on urban heat island in india. Scientific Reports, 7(1):14054, 2017.

Mark G Lawrence. The Relationship between Relative Humidity and the Dew- point Temperature in Moist Air: A Simple Conversion and Applications. Bul- letin of the American Meteorological Society, 86(2):225–233, February 2005.

Tom K R Matthews, Robert L Wilby, and Conor Murphy. Communicating the deadly consequences of global warming for human heat stress. Proceedings of the National Academy of Sciences, 114(15):3861–3866, April 2017.

[Figure]

Brigitte Mueller and Sonia I Seneviratne. Hot days induced by precipitation deficits at the global scale. Proceedings of the National Academy of Sciences of the United States of America, 109(31):12398–12403, July 2012.
 Steven C Sherwood and Matthew Huber. An adaptability limit to climate change due to heat stress. Proceedings of the National Academy of Sciences, 107(21):9552–9555, May 2010.
 Katharine M Willett and Steven Sherwood. Exceedance of heat index thresholds for 15 regions under a warming climate using the wet-bulb globe temperature. International journal of climatology, 32(2):161–177, December 2010.
 Ian N Williams, Raymond T Pierrehumbert, and Matthew Huber. Global warm- ing, convective threshold and false thermostats. Geophysical Research Letters, 36(21):L21805, November 2009.

Ian N Williams and Raymond T Pierrehumbert. Observational evidence against strongly stabilizing tropical cloud feedbacks. Geophysical Research Letters, 44(3):1503–1510, 2017.

Ryan A Zamora, Robert L Korty, and Matthew Huber. Thermal Stratification in Simulations of Warm Climates: A Climatology Using Saturation Potential Vorticity. J. Climate, 29(14):5083–5102, July 2016.

Please also note the supplement to this comment:
https://www.atmos-chem-phys-discuss.net/acp-2019-109/acp-2019-109-RC1-supplement.zip
* * *
[Figure]

[Figure]

**Fig. 1.** (*A*) Histograms of 2-meter $T$ (*Black*), $T_{max}$ (*Blue*), and $T_{W(max)}$ (*Red*) on land from 60S–60N during the last decade (1999–2008). "Max" histograms are annual maxima accumulated over location and year, while the $T$ histogram is accumulated over location and reanalysis time. Data are from the ERA-Interim reanalysis 4xdaily product (similar results are found for the 50m level from the NCEP reanalysis, see *SI Text*). (*B*) Map of $T_{W(max)}$. (*C* and *D*) Same as *A* and *B* but from a slab-ocean version of the CAM3 climate model that produces global-mean surface temperature close to modern values. (*E* and *F*) Same as *C* and *D* but from a high-$CO_2$ model run that produces a global-mean $T$ 12 °C warmer; accounting for GCM bias, the $T_{W(max)}$ distributions are roughly what would be expected with 10 °C of global-mean warming relative to the last decade (see text). Dashed line in *E* is $T_{W(max)}$ reproduced from *C*. White land areas in *F* exceed 35 °C.

**Fig. 1.** Sherwood and Huber figure

$$\overline{\overline{\mathrm{HI}}} = a + b\overline{\overline{T}} + c\overline{\overline{\mathrm{RH}}} + d\overline{\overline{T\mathrm{RH}}} + e\overline{\overline{T^2}} + f\overline{\overline{\mathrm{RH}^2}}$$

$$+ g\overline{\overline{T^2\mathrm{RH}}} + h\overline{\overline{T\mathrm{RH}^2}} + i\overline{\overline{T^2\mathrm{RH}^2}}$$

$$+ \left[ \overline{d\mathrm{RH}'T'} + \overline{eT'^2} + \overline{f\mathrm{RH}'^2} + \overline{gT'^2\mathrm{RH}'} \right.$$

$$\left. + \overline{hT'\mathrm{RH}'^2} + \overline{iT'^2\mathrm{RH}'^2} \right], \qquad (13)$$

where $a$, $b$, $c$, $d$, $e$, $f$, $g$, $h$, and $i$ are constants in the polynomial. RH and $T$ are relative humidity and temperature, respectively. We are not concerned with the terms outside the brackets, as they are the means. The terms within the bracket are representative of turbulent effects on the heat index, which we are discussing. It is these turbulent states

**Fig. 2.** Buzan2015_Eq.13

[Figure]

**Figure 1.** TOA cloud LW flux as a function of (a) SST and
(b) $s_{diff}$; TOA cloud SW flux as a function of (c) SST and
(d) $s_{diff}$; Solid blue and dashed red lines correspond to the
ensemble median over years $0-20$ and $60-80$, respectively,
from 15 IPCC AR4 coupled ocean-atmosphere models for
the 1% per year scenario. Vertical lines indicate the
interquartile range.

**Fig. 3.** Williams_2009_Moist_Convection_Limits_Theory

[Figure]

**Figure 2.** Effects of tropics-wide SST variability on observed and modeled CRE, shown as functions of local SST (Figures 2a, 2c, 2e, and 2g) and *B* (Figures 2b, 2d, 2f, and 2h). (a) The SST distributions of (minus) shortwave (dashed lines) and longwave (solid lines) CRE shift almost uniformly to warmer SSTs during the 2015–2016 El Niño relative to the three coldest years observed. (b) The buoyancy distribution of CRE is approximately invariant with warming. (c) Relative frequency distributions show that warming is approximately uniform across SST. (d) Warming results in little change to the relative frequency distribution of buoyancy. (e–h) As in Figures 2a–2d but for CAM5-AMIP simulations, for the three warmest and coldest simulated years. Error bars indicate the range over all three years.

**Fig. 4.** Williams_2017_Moist_Convection_Limits_Observation

---

## Author Comment (AC1) · 4 Jun 2019

First of all, we would like to thank the anonymous reviewer for the detailed and rich bibliography return.

The general remark is the non-consideration of temperature-humidity covariance via a pure thermodynamical variable.

1st general comment of reviewer #1: "the authors do not use the correct definition of thermodynamics in their analysis" It is true that our study focused in part on the temperature and humidity, as well as on the precipitation trends observed in the Paris

area. There are many thermodynamical variables in the bibliography proposed by the reviewer #1. Most of them are based on different comfort algorithms as detailed in Buzan et al., 2015. These variables give a better indication of human heat stress, hence the terms "thermal comfort" or "feel-like" temperature are used, as highlighted by Matthews et al., 2017. The majority of these indicators, such as HI (Heat Index), HUMIDEX or Tw (Wet Bulb Temperature), use both temperature and relative humidity and are based on risk levels determined by thresholds. In this study, our objective is not to characterize heat stress via a purely thermodynamic variable but to characterize the part of the changes in temperature and precipitation that are related to thermodynamical processes, i.e all processes which modify the content of heat and moisture of the atmosphere but large-scale advection (through surface heat and radiative fluxes, phase changes, radiative effects of particles, Clausius-Clapeyron equation...). The partitioning method used in the manuscript to determine the dynamical and thermodynamical contributions of the trend is widely used (Cassano et al., 2007; Horton et al., 2015; Screen, 2017; Uotila et al., 2007). This method assumes that each weather regime is stationary in time, which is probably not perfect. Hence, the dynamical contribution corresponds to the changes in the occurrence frequency of each circulation pattern, assuming that the circulation patterns are the same during the two periods (but they have been computed overall years covering at least the two periods so that the differences between the two periods are minimized). The thermodynamical contribution inside a weather regime is the result of influences unrelated to circulation, such as changes in long-wave radiation from increasing greenhouse gas concentrations or different cloud macro and microphysics properties, or changes in surface fluxes of moisture and/or radiation. The third component represents the interaction between dynamic and thermodynamic changes, and captures contributions that result from changes in the dynamical component acting on changes in the thermodynamical component. To better understand the dynamical and thermodynamical terms used here, I will add a paragraph similar to the one above in the manuscript.

Summing the thermodynamical (dynamical) change over all weather regimes gives

the total thermodynamical (dynamical) change. For example, the observed trend in summer precipitation (Figure 1) result from thermodynamical changes to 67.8% such as radiation, surface fluxes or moisture change as well as dynamic changes to 32.5% (occurrence of weather regimes). Together, these results suggest that the observed increase in summer precipitation is attributable to both increasing frequency of NAO-weather regimes and changes in the surface water and energy balance. The first version of the manuscript omits the residual term in the contribution tables. In the new version, an additional column will be added, see Figure 1 and 2 below.

2nd general comment of reviewer #1: "Temperature and humidity are analyzed independently, when there is well established literature demonstrating these variables are co-dependent. Temperature and humidity covary together, and non-linearly in extreme regimes. [. . .] I cannot determine if this was taken into account."

Indeed, in this paper, temperature and relative humidity are measured and analyzed independently. The specific humidity q, is computed as a thermodynamical variable based on temperature and relative humidity via the formula below:

q= (0.622* p_sat(T)*RH)/(101325- p_sat(T)*RH)

With p_sat(T)=exp[23.3265- 3802.7/T- (472.68/T)ˆ2 ]

RH: relative humidity from 0 to 1

T: temperature in Kelvin

psat(T): saturated vapour pressure in Pascal

To address the question of Reviewer #1, we completed our analysis by computing the Wet bulb temperature (Tw) based on the formulation of Davies-Jones, 2008 as advised by the reviewer. The figures below present the analysis of Tw at the Montsouris station because the pressure is required for Tw estimations, and it is available since 1979 only for Montsouris, so we can't do this analysis with the other stations.

From a seasonal analysis (Figure 3) no trend is significant for Tw, unlike T2m. In Summer (JJA), although the PDFs of T2m (in black) present changes in the extreme values, the PDFs of Tw, are very similar especially since the decrease in relative humidity compensate the increase in temperature, causing little change in heat stress. The same characteristics are observed by classifying the summer season into four weather regimes (Figure 4).

The reviewer proposed to change Figure 13 of the manuscript to only represent Tw and PRCP (Figure 5). Although Tw is interesting to analyze we think that Tw does not really reflect our objective, which is relative to the understanding of the modification of the local water cycle, especially the presence of a possible surface drying which will impact the formation of clouds and precipitation. For such topic, relative and specific humidity are better adapted. The other reason is that precipitation depends on temperature and humidity and we need to have these two informations independently.

Figure 6 shows the seasonal averages of the T2m/q relationship and the T2m/Tw relationship (same as figure 14 in the paper but exclusively for Montsouris). Very similar patterns between q and Tw supports the idea that q plays the role of thermodynamic variable without necessarily needing information on heat stress.

The reviewer suggests to use the "maximum wet bulb temperature" used by Sherwood and Huber, 2010 on figure 13 of the manuscript. They calculated Twmax histograms as the annual maxima accumulated over the globe (ERA-Interim grid) and year (1999-2008). In our case if we apply the same method we would have a PDF1979-2002 built with only 24 points (one location, 24 years) and a second PDF2003-2017 built with 15 points. This is a very unrepresentative sample to plot a distribution.

Regarding the calculation of "heat stress", in addition to HI, Diffenbaugh et al., 2007, also use Tmax and Tmin. Mueller and Seneviratne, 2012, who show that surface moisture deficits are a relevant factor for the occurrence of hot extremes, define Tmax over the 90th percentile. In our paper we don't use any co-dependent variables but

Tmax and Tmin are used for extreme index calculations, giving a first indication of the trend of the thermal extremes.

In our paper we focus on the observed trends and we want to keep the independent analysis between temperature and relative humidity, because this surface drying can play a major role in the trend of other variables such as turbulent flows, and thus can intensify or inhibit existing surface-atmosphere feedbacks. Specific humidity allows to account for the link between temperature & humidity. As thermal comfort is not the main object of the article and do not bring very different information compares to specific and relative humidity, the choice was made not to add information on thermal comfort in the article.

The perspective of this study is to use SIRTA supersite (near Paris) which measures more meteorological variables at hourly resolution since 2003, in order to identify the processes explaining the trends and to improve our knowledge on these surface-atmosphere processes at the local scale.

Buzan, J. R., Oleson, K. and Huber, M.: Implementation and comparison of a suite of heat stress metrics within the Community Land Model version 4.5, Geosci. Model Dev., 8(2), 151–170, doi:10.5194/gmd-8-151-2015, 2015.

Cassano, J. J., Uotila, P., Lynch, A. H. and Cassano, E. N.: Predicted changes in synoptic forcing of net precipitation in large Arctic river basins during the 21st century, J. Geophys. Res. Biogeosciences, 112(G4), n/a-n/a, doi:10.1029/2006JG000332, 2007.

Davies-Jones, R.: An Efficient and Accurate Method for Computing the Wet-Bulb Temperature along Pseudoadiabats, Mon. Weather Rev., 136(7), 2764–2785, doi:10.1175/2007MWR2224.1, 2008.

Diffenbaugh, N. S., Pal, J. S., Giorgi, F. and Gao, X.: Heat stress intensification in the Mediterranean climate change hotspot, Geophys. Res. Lett., 34(11), doi:10.1029/2007GL030000, 2007.

Horton, D. E., Johnson, N. C., Singh, D., Swain, D. L., Rajaratnam, B. and Diffenbaugh, N. S.: Contribution of changes in atmospheric circulation patterns to extreme temperature trends, Nature, 522, 465, 2015.

Matthews, T. K. R., Wilby, R. L. and Murphy, C.: Communicating the deadly consequences of global warming for human heat stress, Proc. Natl. Acad. Sci., 114(15), 3861–3866, doi:10.1073/pnas.1617526114, 2017.

Mueller, B. and Seneviratne, S. I.: Hot days induced by precipitation deficits at the global scale, Proc. Natl. Acad. Sci., 109(31), 12398–12403, doi:10.1073/pnas.1204330109, 2012.

Screen, J. A.: The missing Northern European winter cooling response to Arctic sea ice loss, Nat. Commun., 8, 14603, 2017.

Sherwood, S. C. and Huber, M.: An adaptability limit to climate change due to heat stress, Proc. Natl. Acad. Sci., 107(21), 9552–9555, doi:10.1073/pnas.0913352107, 2010.

Uotila, P., Lynch, A. H., Cassano, J. J. and Cullather, R. I.: Changes in Antarctic net precipitation in the 21st century based on Intergovernmental Panel on Climate Change (IPCC) model scenarios, J. Geophys. Res. Atmospheres, 112(D10), doi:10.1029/2006JD007482, 2007.

[Figure]

| | Dynamical contribution [mm (%)] | Thermodynamical contribution [mm (%)] | Residual term [mm (%)] | $\Delta PRCP$ [mm] |
|---|---|---|---|---|
| SUMMER (JJA) | 5.32 (**32.5**) | 11.10 (**67.8**) | -0.04 (**-0.3**) | 16.38 |
| | Dynamical contribution [mm] | Thermodynamical contribution [mm] | Residual term [mm] | $\Delta PRCP_i$ [mm] |
| NAO- | 20.39 | -2.85 | -0.02 | 17.53 |
| Atlantic Ridge | -10.27 | 3.57 | -0.01 | -6.71 |
| Blocking | -8.70 | 9.47 | -0.02 | 0.75 |
| Atlantic Low | 3.90 | 0.90 | 0 | 4.81 |

**Fig. 1.** Dynamical, thermodynamical and residual contributions of the precipitation change ($\Delta$PRCP) in mm for summer (JJA) and for the four weather regimes in summer. Values in parenthesis give the ratio (in %)

|  | Dynamical contribution [°C (%)] | Thermodynamical contribution [°C (%)] | Residual term [°C (%)] | $\Delta T$ [°C] |
|---|---|---|---|---|
| WINTER (DJF) | 0.06 (*29.6*) | 0.17 (*78.9*) | -0.02 (*-8.5*) | 0.21 |
| SUMMER (JJA) | -0.05 (*-5.9*) | 0.87 (*103.2*) | 0.02 (*2.7*) | 0.84 |

**Fig. 2.** Dynamical, thermodynamical and residual contributions of the temperature change ($\Delta$T) in °C in winter (DJF) and in summer (JJA). Values in parenthesis give the ratio (in %) between the change component

[Figure]

**Fig. 3.** Left: Mann Kendall seasonal trends for T2m in black and Tw in blue. The red value represents the Sen slope in units per decade. A solid bar indicates a significant trend for a confidence interval of p

[Figure]

**Fig. 4.** Same as Figure 3 but for summer weather regimes

[Figure]

**Fig. 5.** Violin plot of daily Tw (first line) and PRCP (second line) for the four weather regimes between the periods 1979-2002 and 2003-2017. Box numbers represent trends in unit decade-1 over the period 1979

**Fig. 6.** T2m – q seasonal relationship in Montsouris in filled circle, and T2m – Tw seasonal relationship in Montsouris in empty triangle. Each point represents the seasonal average of one years.

---

## Referee Comment (RC2) · Anonymous Referee #3 · 12 Jul 2019

The authors examine the variability of temperature, moisture, and precipitation of local scales at different time scales using local observations during 1979 to 2017, and they try to determine the contributions of thermodynamic and dynamical processes to the observed extremes. Although it is a good topic, which is worth publication in ACP, I have a few comments and concerns about the manuscript. I recommend publication after the concerns and comments are successfully addressed.

Major comments: 1) Since the temperature and moisture are co-dependent and they vary together. it seems to me that it is not correct enough to discuss them separately. Therefore, the authors may not discuss the thermodynamic contributions using the

correct definition. 2) The methodology section is so lengthy that bury the effective information. I would like to suggest to rewritten this section to make your key method more clear.

Minor comments: 1. The classification of four seasons appear twice in the manuscript, one is around Page 3, Line 23, the other one is around Page 9, Line 43. 2. Given the uncertainty of your calculated Kendall Tau, the differences among these $\tau$s may not significantly. Could you please provide the CI of your calculated Kendall Tau? 3. The short citation in the text is not in good format. There should be parentheses covering the year. For instance, Donat et al., 2013 (Page 2, Line 1) should be Donat et al. (2013). 4. Page 5, Line 19, "with d ÐĎ [1 à 365]" could be a symbol issue. 5. For those tables, horizontal lines should normally only appear above and below the table. 6. Figure 11: please add the meaning of each horizontal lines into the figure caption.

---

## Author Comment (AC2) · 17 Jul 2019

We want to thank the reviewer for these relevant comments. You'll find below our answer to each of your remarks

Major comments:

1) Since the temperature and moisture are codependent and they vary together. it seems to me that it is not correct enough to discuss them separately. Therefore, the authors may not discuss the thermodynamic contributions using the correct definition.

In this paper, temperature and relative humidity are measured and analyzed independently. That's why the specific humidity q, is computed as a thermodynamical variable based on temperature and relative humidity via the formula below:

q= (0.622* p_sat(T)*RH)/(101325- p_sat(T)*RH)

With p_sat(T)=exp[23.3265- 3802.7/T- (472.68/T)ˆ2 ]

RH: relative humidity from 0 to 1 T: temperature in Kelvin psat(T): saturated vapour pressure in Pascal

Theoretically, with global warming, the rise of temperature should be accompanied by an increase of the specific humidity at constant relative humidity. However, at the Paris scale, the increase of the water holding capacity of the atmosphere (associated with the temperature increase) is not accompanied by an increase of surface humidity (q) leading to a decrease of surface relative humidity. In our paper we focus on the observed trends and we want to keep the independent analysis between temperature and relative humidity, because this surface drying can play a major role in the trend of other variables such as turbulent flows, and thus can intensify or inhibit existing surface-atmosphere feedbacks. However, specific humidity allows to account for the link between temperature & humidity. Moreover, as an answer to Reviewer Comment 1, we completed our analysis by computing the Wet bulb temperature (Tw) based on the formulation of Davies-Jones, 2008 . As the specific humidity, Tw keeps the codependence between temperature and moisture. Figure 1. below shows the seasonal averages of the T2m/q relationship and the T2m/Tw relationship (same as figure 14 in the paper but exclusively for Montsouris). Very similar patterns between q and Tw supports the idea that q plays the role of thermodynamic variable without necessarily needing information on heat stress. As thermal comfort is not the main object of the article and do not bring very different information compares to specific, the choice was made not to add information on thermal comfort in the article.

The comments of the reviewer may actually refer to the estimation of the dynamical and thermodynamical contributions of the observed changes. Indeed, such terms are

commonly used in the literature with the same approach that we used (Cassano et al., 2007; Horton et al., 2015; Screen, 2017; Uotila et al., 2007)

2) The methodology section is so lengthy that bury the effective information. I would like to suggest to rewritten this section to make your key method more clear.

Following this comment, we deleted the part "3.3 Statistical characteristics of the PDFs" in methodology and the coefficients associated in Figures 4 and 6, to reduce information. We moved the paragraph "3.4 Climate indices" in data part especially in the part "2.1. Observation". We also made some changes within the remaining sections: see below the new paragraph: "3. Methodology".

In supplement the rewritten methodology part.

Minor comments:

3) The classification of four seasons appear twice in the manuscript, one is around Page 3, Line 23, the other one is around Page 9, Line 43.

Thank you for this observation, we keep the details of the seasons at the first appearance Page 3, Line 23.

4) Given the uncertainty of your calculated Kendall Tau, the differences among these tau's may not significantly. Could you please provide the CI of your calculated Kendall Tau?

Kendall tau is a rank coefficient. It is calculated as:

Tau=((concordant pairs)-(discordant pairs))/(number of pair combinations)

That is to say that in each pair of points if the trend increases then there is a concordance (+1) inversely if the trend decreases there is discordance (-1). In this paper the significance test is performed on the Sen Slope. So we can't give the confidence interval of the calculated Kendall Tau.

5) The short citation in the text is not in good format. There should be parentheses covering the year. For instance, Donat et al., 2013 (Page 2, Line 1) should be Donat et al. (2013).

This is now corrected in all the paper

6) Page 5, Line 19, "with d ÃŘĚĞD [1 à 365]" could be a symbol issue.

There must be a problem of encoding in pdf. Now it's: "with d, ranging from 1 to 365"

7) For those tables, horizontal lines should normally only appear above and below the table.

This is now changed in all the paper

8) Figure 11: please add the meaning of each horizontal lines into the figure caption.

Corrected: this is the new legend for the Figure 11: "Figure 11: "Summer Days" frequency (Tmax >25°C) in number of days for the JJA season (black boxplot) and for each summer weather regime calculated over the period 1979-2017. The bottom and top edges of the box indicate the 25th and 75th percentiles, respectively and the central line the median. The bottom and top lines outside the box indicate the minimal and maximal values respectively."

Cassano, J. J., Uotila, P., Lynch, A. H. and Cassano, E. N.: Predicted changes in synoptic forcing of net precipitation in large Arctic river basins during the 21st century, J. Geophys. Res. Biogeosciences, 112(G4), n/a-n/a, doi:10.1029/2006JG000332, 2007.

Davies-Jones, R.: An Efficient and Accurate Method for Computing the Wet-Bulb Temperature along Pseudoadiabats, Mon. Weather Rev., 136(7), 2764–2785, doi:10.1175/2007MWR2224.1, 2008.

Horton, D. E., Johnson, N. C., Singh, D., Swain, D. L., Rajaratnam, B. and Diffenbaugh, N. S.: Contribution of changes in atmospheric circulation patterns to extreme temperature trends, Nature, 522, 465, 2015.

Screen, J. A.: The missing Northern European winter cooling response to Arctic sea ice loss, Nat. Commun., 8, 14603, 2017.

Uotila, P., Lynch, A. H., Cassano, J. J. and Cullather, R. I.: Changes in Antarctic net precipitation in the 21st century based on Intergovernmental Panel on Climate Change (IPCC) model scenarios, J. Geophys. Res. Atmospheres, 112(D10), doi:10.1029/2006JD007482, 2007.

Please also note the supplement to this comment:
https://www.atmos-chem-phys-discuss.net/acp-2019-109/acp-2019-109-AC2-supplement.pdf

[Figure]

[Figure]

**Fig. 1.** T2m – q seasonal relationship in Montsouris in filled circle, and T2m – Tw seasonal relationship in Montsouris in empty triangle. 
[revised manuscript text omitted]

---

## Author Response (AR1)

**Response to Referee#1**

First of all, we would like to thank the anonymous reviewer for the detailed and rich bibliography return.

5   The general remark is the non-consideration of temperature-humidity covariance via a pure thermodynamical variable.

**RC1-1:** **1st general comment is: "***the authors do not use the correct definition of thermodynamics in their analysis***"**

It is true that our study focused in part on the temperature and humidity, as well as on the precipitation trends observed in the Paris area. There are many thermodynamical variables in the bibliography proposed by the referee#1. Most of them are based on different comfort algorithms as detailed in Buzan et al. (2015). These variables give a better indication of human heat stress, hence the terms "thermal comfort" or "feel-like" temperature are used,

15   as highlighted by Matthews et al. (2017). The majority of these indicators, such as HI (Heat Index), HUMIDEX or Tw (Wet Bulb Temperature), use both temperature and relative humidity and are based on risk levels determined by thresholds. In this study, our objective is not to characterize heat stress via a purely thermodynamic variable but to characterize the part of the changes in temperature and precipitation that are related to thermodynamical processes, i.e all the processes which modify the content of heat and moisture of the atmosphere but large-scale

20   advection (through surface heat and radiative fluxes, phase changes, radiative effects of particles, Clausius-Clapeyron equation…).

To better understand what we mean when we say dynamic and thermodynamic term, we added a paragraph Page 23, lines 12 to 21 in the track changes version below:

*"The partitioning method used in the manuscript to determine the dynamical and thermodynamical contributions of the trend is widely used (Cassano et al., 2007; Uotila et al., 2007; Horton et al., 2015; Screen, 2017). This method assumes that each weather regime is stationary in time, which is probably not perfect. Hence, the dynamical contribution corresponds to the changes in the occurrence frequency of each*

30   *circulation pattern, assuming that the circulation patterns are the same during the two periods (but they have been computed over all years covering at least the two periods so that the differences between the two periods are minimized). The thermodynamical contribution inside a weather regime is the result of influences unrelated to circulation, such as changes in long-wave radiation from increasing greenhouse gas concentrations or different cloud macro- and micro-physical properties, or changes in surface fluxes of*

35   *moisture and/or radiation. The third component represents the interaction between dynamic and thermodynamic changes, and captures contributions that result from changes in the dynamical component acting on changes in the thermodynamical component."*

The first version of the manuscript omits the residual term in the contribution tables. In the new version, an
40   additional column is added, see the new Table 2 and Table 3; and some modifications about values have been corrected for example Page 23, Line 23

**RC1-2: 2nd general comment is:** *"Temperature and humidity are analyzed independently, when there is well established literature demonstrating these variables are co-dependent. Temperature and humidity covary together, and non-linearly in extreme regimes. [...] I cannot determine if this was taken into account."*

5   Indeed, in this paper, temperature and relative humidity are measured and analyzed independently. The specific humidity $q$, is computed as a thermodynamical variable based on temperature and relative humidity via the formula below:

$$q = \frac{0.622 * p_{sat}(T) * RH}{101325 - p_{sat}(T) * RH}$$

With      $p_{sat}(T) = exp\left[23.3265 - \frac{3802.7}{T} - \left(\frac{472.68}{T}\right)^2\right]$

RH: relative humidity from 0 to 1
T: temperature in Kelvin
$p_{sat}$(T): saturated vapour pressure in Pascal

To address the question of referee#1, we completed our analysis by computing the Wet bulb temperature (Tw) based on the formulation of Davies-Jones (2008) as advised by the referee#1. The figures below present the analysis of Tw at the Montsouris station because the pressure is required for Tw estimations, and it is available since 1979
20   only for Montsouris, so we can't do this analysis with the other stations. From a seasonal analysis (Figure A) no trend is significant for Tw, unlike T2m. In Summer (JJA), although the PDFs of T2m (in black) present changes in the extreme values, the PDFs of Tw do not evolve because the decrease in relative humidity compensates the increase in temperature, causing little change in heat stress.

25   The same characteristics are observed by classifying the summer season into four weather regimes (Figure B).

The referee#1 proposed to change Figure 13 of the manuscript to only represent Tw and PRCP (Figure C). Although Tw is interesting to analyze we think that Tw does not really reflect our objective, which is relative to the understanding of the modification of the local water cycle, especially the presence of a possible surface drying
30   which will impact the formation of clouds and precipitation. For such topic, relative and specific humidity are better adapted. The other reason is that precipitation depends on temperature and humidity and we need to have these two informations independently.

Figure D shows the seasonal averages of the T2m/q relationship and the T2m/Tw relationship (same as figure 14
35   in the paper but exclusively for Montsouris). Very similar patterns between $q$ and Tw supports the idea that $q$ plays the role of thermodynamic variable without necessarily needing information on heat stress.

The referee#1 suggests to use the "maximum wet bulb temperature" used by Sherwood and Huber (2010) on figure 13 of the manuscript. They calculated $Tw_{max}$ histograms as the annual maxima accumulated over the globe (ERA-
40   Interim grid) and year (1999-2008). In our case if we apply the same method we would have a $PDF_{1979-2002}$ built with only 24 points (one location, 24 years) and a second $PDF_{2003-2017}$ built with 15 points. This sampling size is not representative enough to plot a distribution.

Regarding the calculation of "heat stress", in addition to HI, Diffenbaugh et al. (2007) also use Tmax and Tmin. Mueller and Seneviratne (2012), who show that surface moisture deficits are a relevant factor for the occurrence of hot extremes, define Tmax over the 90th percentile. In our paper we don't use any co-dependent variables but Tmax and Tmin are used for extreme index calculations, giving a first indication of the trend of the thermal extremes.

In our paper we focus on the observed trends and we want to keep the independent analysis between temperature and relative humidity, because this surface drying can play a major role in the trend of other variables such as turbulent flows, and thus can intensify or inhibit existing surface-atmosphere feedbacks. Specific humidity allows to account for the link between temperature & humidity. As thermal comfort is not the main object of the article and do not bring very different information compares to specific and relative humidity, the choice was made not to add information on thermal comfort in the article.

The perspective of this study is to use SIRTA supersite (near Paris) which measures more meteorological variables at hourly resolution since 2003, in order to identify the processes explaining the trends and to improve our knowledge on these surface-atmosphere processes at the local scale.

We added a paragraph (below) in the discussion, on the choice to independently analyze temperature and relative humidity, and the choice to keep specific humidity as a co-dependent variable. Page 29, lines 4 to 14 in the track changes version:

*"Theoretically, with global warming, the rise of temperature should be accompanied by an increase of the specific humidity for a given relative humidity. At Paris scale, the increase in the water retention capacity of the atmosphere (related to the increase in temperature) is not accompanied by an increase in the surface moisture (q), which leads to a decrease in surface relative humidity. There are several thermodynamical variables that take into account this co-dependence between temperature and relative humidity, which is often used to calculate heat stress (Buzan et al., 2015; Davies-Jones, 2008; Sherwood and Huber, 2010; Willet and Sherwood, 2012). We observed that the evolution on the specific humidity is similar to other coupled temperature and humidity variables, as the wet bulb temperature calculated via the formula of Davies-Jones (2008), meaning that the heat stress is constant (temperature increases but relative humidity decreases). In this paper, we use the specific humidity as one of these co-dependent variables. We then observe a surface drying, which can play a major role in the trend of other variables such as turbulent flows, and thus can intensify or inhibit existing surface-atmosphere feedbacks."*

[Figure]

**Figure A.** *Left: Mann Kendall seasonal trends for T2m in black and Tw in blue. The red value represents the Sen slope in units per decade. A solid bar indicates a significant trend for a confidence interval of p=0.05, and a mosaic bar indicates a non-significant trend. Right: Seasonal PDF of the daily anomalies of T2m in black and Tw in blue. Dashed lines for 1979-2002 period and solid lines for the 2003-2017 period. Anomalies are compute over the period 1979-2017.*

[Figure]

**Figure B.** *Same as Figure 1 but for summer weather regimes.*

[Figure]

**Figure C.** *Violin plot of daily Tw (first line) and PRCP (second line) for the four weather regimes between the periods 1979-2002 and 2003-2017. Box numbers represent trends in unit decade$^{-1}$ over the period 1979-2017.*

[Figure]

**Figure D.** *T2m – q seasonal relationship in Montsouris in filled circle, and T2m – Tw seasonal relationship in Montsouris in empty triangle. Each point represents the seasonal average of one years.*

Buzan, J. R., Oleson, K. and Huber, M.: Implementation and comparison of a suite of heat stress metrics within the Community Land Model version 4.5, Geosci. Model Dev., 8(2), 151–170, doi:10.5194/gmd-8-151-2015, 2015

5   Cassano, J. J., Uotila, P., Lynch, A. H. and Cassano, E. N.: Predicted changes in synoptic forcing of net precipitation in large Arctic river basins during the 21st century, J. Geophys. Res. Biogeosciences, 112(G4), n/a-n/a, doi:10.1029/2006JG000332, 2007

Davies-Jones, R.: An Efficient and Accurate Method for Computing the Wet-Bulb Temperature along Pseudoadiabats, Mon. Weather Rev., 136(7), 2764–2785, doi:10.1175/2007MWR2224.1, 2008

Diffenbaugh, N. S., Pal, J. S., Giorgi, F. and Gao, X.: Heat stress intensification in the Mediterranean climate change hotspot, Geophys. Res. Lett., 34(11), doi:10.1029/2007GL030000, 2007

10  Horton, D. E., Johnson, N. C., Singh, D., Swain, D. L., Rajaratnam, B. and Diffenbaugh, N. S.: Contribution of changes in atmospheric circulation patterns to extreme temperature trends, Nature, 522, 465, 2015

Matthews, T. K. R., Wilby, R. L. and Murphy, C.: Communicating the deadly consequences of global warming for human heat stress, Proc. Natl. Acad. Sci., 114(15), 3861–3866, doi:10.1073/pnas.1617526114, 2017

15  Mueller, B. and Seneviratne, S. I.: Hot days induced by precipitation deficits at the global scale, Proc. Natl. Acad. Sci., 109(31), 12398–12403, doi:10.1073/pnas.1204330109, 2012

Screen, J. A.: The missing Northern European winter cooling response to Arctic sea ice loss, Nat. Commun., 8, 14603, 2017

Sherwood, S. C. and Huber, M.: An adaptability limit to climate change due to heat stress, Proc. Natl. Acad. Sci., 107(21), 9552–9555, doi:10.1073/pnas.0913352107, 2010

20  Uotila, P., Lynch, A. H., Cassano, J. J. and Cullather, R. I.: Changes in Antarctic net precipitation in the 21st century based on Intergovernmental Panel on Climate Change (IPCC) model scenarios, J. Geophys. Res. Atmospheres, 112(D10), doi:10.1029/2006JD007482, 2007

Willett, K. M. and Sherwood, S.: Exceedance of heat index thresholds for 15 regions under a warming climate using the wet-bulb globe temperature, Int. J. Climatol., 32(2), 161–177, doi:10.1002/joc.2257, 2012.

**Response to Referee#3**

We want to thank the reviewer for these relevant comments. You'll find below our answer to each of your remarks

5 **Major comments:**

**RC3-1: Since the temperature and moisture are codependent and they vary together. it seems to me that it is not correct enough to discuss them separately. Therefore, the authors may not discuss the thermodynamic contributions using the correct definition.**

In this paper, temperature and relative humidity are measured and analyzed independently. That's why the specific humidity $q$, is computed as a thermodynamical variable based on temperature and relative humidity via the formula below:

$$q = \frac{0.622 * p_{sat}(T) * RH}{101325 - p_{sat}(T) * RH}$$

With $\quad p_{sat}(T) = exp\left[23.3265 - \frac{3802.7}{T} - \left(\frac{472.68}{T}\right)^2\right]$

RH: relative humidity from 0 to 1
T: temperature in Kelvin
$p_{sat}$(T): saturated vapour pressure in Pascal

25 Theoretically, with global warming, the rise of temperature should be accompanied by an increase of the specific humidity at constant relative humidity. However, at the Paris scale, the increase of the water holding capacity of the atmosphere (associated with the temperature increase) is not accompanied by an increase of surface humidity (q) leading to a decrease of surface relative humidity.
In our paper we focus on the observed trends and we want to keep the independent analysis between temperature

30 and relative humidity, because this surface drying can play a major role in the trend of other variables such as turbulent flows, and thus can intensify or inhibit existing surface-atmosphere feedbacks. However, specific humidity allows to account for the link between temperature & humidity.
Moreover, as an answer to referee#1, we completed our analysis by computing the Wet bulb temperature (Tw) based on the formulation of of Davies-Jones (2008). As the specific humidity, Tw keeps the codependence between

35 temperature and moisture.
Figure A below shows the seasonal averages of the T2m/q relationship and the T2m/Tw relationship (same as figure 14 in the paper but exclusively for Montsouris). Very similar patterns between $q$ and Tw supports the idea that $q$ plays the role of thermodynamic variable without necessarily needing information on heat stress. As thermal comfort is not the main object of the article and do not bring very different information compares to specific, the

40 choice was made not to add information on thermal comfort in the article.

[Figure]

Figure A. *T2m – q seasonal relationship in Montsouris in filled circle, and T2m – Tw seasonal relationship in Montsouris in empty triangle. Each point represents the seasonal average of one years.*

The comments of the referee#3 may actually refer to the estimation of the dynamical and thermodynamical contributions of the observed changes. Indeed, such terms are commonly used in the literature with the same approach that we used (Cassano et al., 2007; Uotila et al., 2007; Horton et al., 2015; Screen, 2017).
To better understand what we mean when we say dynamic and thermodynamic term, we added a paragraph Page 23, lines 12 to 21 in the track changes version below:

*"The partitioning method used in the manuscript to determine the dynamical and thermodynamical contributions of the trend is widely used (Cassano et al., 2007; Uotila et al., 2007; Horton et al., 2015; Screen, 2017). This method assumes that each weather regime is stationary in time, which is probably not perfect. Hence, the dynamical contribution corresponds to the changes in the occurrence frequency of each circulation pattern, assuming that the circulation patterns are the same during the two periods (but they have been computed over all years covering at least the two periods so that the differences between the two periods are minimized). The thermodynamical contribution inside a weather regime is the result of influences unrelated to circulation, such as changes in long-wave radiation from increasing greenhouse gas concentrations or different cloud macro- and micro-physical properties, or changes in surface fluxes of*

*moisture and/or radiation. The third component represents the interaction between dynamic and thermodynamic changes, and captures contributions that result from changes in the dynamical component acting on changes in the thermodynamical component."*

We also added a paragraph in the discussion, on the choice to independently analyze temperature and relative humidity, and the choice to keep specific humidity as a co-dependent variable. Page 29, lines 4 to 14 in the track changes version:

*"Theoretically, with global warming, the rise of temperature should be accompanied by an increase of the specific humidity for a given relative humidity. At Paris scale, the increase in the water retention capacity of the atmosphere (related to the increase in temperature) is not accompanied by an increase in the surface moisture (q), which leads to a decrease in surface relative humidity. There are several thermodynamical variables that take into account this co-dependence between temperature and relative humidity, which is often used to calculate heat stress (Buzan et al., 2015; Davies-Jones, 2008; Sherwood and Huber, 2010; Willet and Sherwood, 2012). We observed that the evolution on the specific humidity is similar to other coupled temperature and humidity variables, as the wet bulb temperature calculated via the formula of Davies-Jones (2008), meaning that the heat stress is constant (temperature increases but relative humidity decreases). In this paper, we use the specific humidity as one of these co-dependent variables. We then observe a surface drying, which can play a major role in the trend of other variables such as turbulent flows, and thus can intensify or inhibit existing surface-atmosphere feedbacks."*

**RC3-2: The methodology section is so lengthy that bury the effective information. I would like to suggest to rewritten this section to make your key method more clear.**

Following this comment, we deleted the part "3.3 Statistical characteristics of the PDFs" in methodology and the coefficients associated in Figures 4 and 6, to reduce information.
We moved the paragraph "3.4 Climate indices" in data part especially in the part "2.1. Observation" (Page 13, lines 14 to 23).
We also made some changes within the remaining sections.
The corrected version of this section appears on Pages 15 to 18 of the track changes version below.

**Minor comments:**

**RC3-3: The classification of four seasons appear twice in the manuscript, one is around Page 3, Line 23, the other one is around Page 9, Line 43.**
Thank you for this observation, we keep the details of the seasons at the first appearance Page 13, line 22 in the track changes version and we have deleted the sentence Page 20, lines 2-4:

"*Our study area is marked by a high seasonal cycle (Fig. 2). For each variable, we apply our analysis for the four seasons as follows: the winter season from December to February (DJF), the spring season from March to May (MAM), the summer season from June to August (JJA) and the autumn season from September to November (SON).*"

**RC3-4: Given the uncertainty of your calculated Kendall Tau, the differences among these τs may not significantly. Could you please provide the CI of your calculated Kendall Tau?**

Kendall tau is a rank coefficient. It is calculated as:

$$\tau = \frac{(concordant\ pairs) - (discordant\ pairs)}{number\ of\ pair\ combinations}$$

That is to say that in each pair of points if the trend increases then there is a concordance (+1) inversely if the trend decreases there is discordance (-1).

In this paper the significance test is performed on the Sen Slope. So we can't give the confidence interval of the calculated Kendall Tau.

**RC3-5: The short citation in the text is not in good format. There should be parentheses covering the year. For instance, Donat et al., 2013 (Page 2, Line 1) should be Donat et al. (2013).**

This is now corrected throughout the paper

**RC3-6: Page 5, Line 19, "with d ĐˇD [1 à 365]" could be a symbol issue.**

There must be a problem of encoding in pdf. Now it's: Page 16, line 4: "with *d,* ranging from 1 to 365"

**RC3-7: For those tables, horizontal lines should normally only appear above and below the table.**

This is now modified in the revised paper

**RC3-8: Figure 11: please add the meaning of each horizontal lines into the figure caption.**

Corrected: this is the new legend for the Figure 11:

[revised manuscript text omitted]

**Commenté [JR2]:** **RC3-2:** This paragraph, previously in the methodology part, has been placed here in the observation data.

**2.2 Comparison of the local observation with the ERA-Interim reanalyses and SAFRAN analysis**

Although the main data sources in this study come from direct observation, it is interesting to test the ability of well-known reanalyses to represent the fine-scale behaviour. To do so, we used the reanalysis from the European Centre for Medium-Range Weather Forecasts (ECMWF) ERA-Interim (Simmons et al., 2014), as well as the high resolution meteorological analysis SAFRAN (Quintana-Seguí et al., 2008), for precipitations. ERA-I shows a general pattern of underestimation of temperatures (T2m, Tmax and Tmin) relative to observations, which is more marked seasonally, especially in spring and summer (Fig. A1b and A1e). In addition, ERA-I also shows a strong overestimation of relative humidity annually and seasonally, whereas the specific humidity is rather well estimated by ERA-I. For precipitation, SAFRAN shows rather

satisfactory results in terms of bias despite the high daily variability (Fig. A1b and A1e). However, the statistical analyses carried out on the monthly accumulations show very good results, confirming that SAFRAN is well adapted to inform the precipitation at local scale, at least for this area. The detailed results obtained from the two datasets are presented in Appendix A. The reasons for the discrepancies between direct observation and ERA-I / SAFRAN are out of the scope of this paper, but the presence of significant bias at this local scale motivates the use of observations and not reanalysis for the current issues.

**2.3 Local climate**

The temporal evolution of the six daily variables, namely, the daily temperature at 2 metres (T2m), the daily maximum temperature at 2 metres (Tmax), the daily minimum temperature at 2 metres (Tmin), the relative humidity (RH), the specific humidity ($q$) and the precipitation (PRCP), on an annual basis and for seasonal scales are presented in Fig. 2. The local climate is characterized by cold and humid winters in contrast to warm and increasingly drier summers. The seasonal averages of T2m and Tmax are similar in spring and autumn; however, autumn has warmer Tmin and wetter conditions than does spring. The relative humidity is the only variable for which the decrease tendency, especially in the spring-summer, clearly appears. Regarding precipitation, the Paris region shows no preferential season when considering the total amount.

**3 Methodology**

**3.1 Mann-Kendall Trend Test**

Trends were calculated using the Mann-Kendall test (Kendall, 1955; Mann, 1945). This test detects the presence of a monotonic tendency in a chronological series of a variable. It is a non-parametric method; that is, it makes no assumptions about the underlying distribution of the data, and its rank-based measure is not influenced by extreme values. This method mainly gives three types of information.

- The Kendall Tau, or Kendall rank correlation coefficient, measures the monotony of the slope. Kendall's Tau varies between -1 and 1; it is positive when the trend increases and vice-versa.
-
- The Sen slope, which estimates the overall slope of the time series. This slope corresponds to the median of all the slopes calculated between each pair of points in the series.
- The significance, which represents the threshold for which the hypothesis that there is no trend is accepted. The trend is statistically significant when the p-value is less than 0.05.

**Commenté [JR3]: RC3-2:** We deleted the part "3.3 Statistical characteristics of the PDFs" in methodology and the coefficients associated in Figures 4 and 6, to reduce information.
We moved the paragraph "3.4 Climate indices" in data part especially in the part "2.1. Observation".
We also made some changes within the remaining sections: see below the new paragraph: "3. Methodology"

[revised manuscript text omitted]

Commenté [JR11]: RC3-2: To reduce methodology part we delete the part "3.3 Statistical characteristics of the PDFs", so the coefficients on the PDF are no longer visible, because indeed, they were little commented in the manuscript.

[Figure]

**Figure 5: Mann-Kendall seasonal trends of temperature climate indices calculated from Météo France observations stations for the four seasons (a) DJF, (b) MAM, (c) JJA, (d) SON. See Table 1 for temperature climate indices. On the abscisse, Kendall's Tau represents the rank correlation coefficient between the variable and time. The red value represents the Sen slope, i.e., the median slope in units per decade, and the black value represents the average original value in 1979 (in unit). A solid bar indicates a significant trend for a confidence interval of $p = 0.05$, and a mosaic bar indicates a non-significant trend.**

[Figure]

**Figure 6: Same as Fig. 4 for RH (a, c, d, e, f) and q (b).**

**Commenté [JR12]:** **RC3-2:** To reduce methodology part we delete the part "3.3 Statistical characteristics of the PDFs", so the coefficients on the PDF are no longer visible, because indeed, they were little commented in the manuscript.

[Figure]

5     **Figure 7: Top, same as Fig. 4a but for PRCP. Bottom, seasonal PDF of daily intensities of rainy days only (> 0.2 mm day⁻¹) for b) DJF, c) MAM, d) JJA and e) SON. Dashed black: the past period from 1979 to 2002; red line: the current period from 2003 to 2017.**

[Figure]

**Figure 8: Same as Fig. 5, but for precipitation climate indices.**

| | Dynamical contribution [mm (*%*)] | Thermodynamical contribution [mm (*%*)] | Residual term [mm (*%*)] | $\Delta PRCP$ [mm] |
|---|---|---|---|---|
| SUMMER (JJA) | 5.32 (***32.5***) | 11.10 (***67.8***) | -0.04 (***-0.3***) | 16.38 |
| NAO- | 20.39 | -2.85 | -0.02 | 17.5 |
| Atlantic Ridge | -10.27 | 3.57 | -0.01 | -6.7 |
| Blocking | -8.70 | 9.47 | -0.02 | 0.8 |
| Atlantic Low | 3.90 | 0.90 | 0 | 4.8 |

**Table 2: Dynamical, thermodynamical and residual contributions of the precipitation change ($\Delta PRCP$) in mm for summer (JJA) and for the four weather regimes in summer. Values in parenthesis give the ratio (in %) between the change components and the total change.**

Commenté [JR13]: **RC1-1:** New table with residual term

**RC3-7:** We delete the lines between summer season and weather regimes.

[Figure]

**Figure 9: Violin plot of daily T2m (first line), RH (second line), $q$ (third line) and PRCP (fourth line) for the four summer weather regimes between the periods 1979-2002 and 2003-2017 (one regime, one column). The black bar represents the mean, and the red bar the median. Boxed numbers represent trends in unit decade$^{-1}$ over the period 1979-2017. The asterisk represents a significant trend for a confidence interval of $p = 0.05$.**

[Figure]

**Figure 10: Mann-Kendall trends in observational data for climate indices for the four summer weather regimes a) NAO-, (b) Atlantic Ridge, (c) Blocking and (d) Atlantic Low. Figure characteristics are the same as for Fig. 7.**

[Figure]

**Figure 11: "Summer Days" frequency (Tmax >25°C) in number of days for the JJA season (black boxplot) and for each summer weather regime calculated over the period 1979-2017. The bottom and top edges of the box indicate the 25th and 75th percentiles, respectively and the central line the median. The bottom and top lines outside the box indicate the minimal and maximal values respectively.**

Commenté [JR14]: RC3-8: the new legend

[Figure]

**Figure 12: Linear trends of SU (in number of day yr⁻¹) as a function of length of segment (y-axis represents the end year of the segment, and x-axis represents the starting year of the segment) for a) Summer, b) NAO-, c) Atlantic Ridge, d) Blocking and e) Atlantic Low. The minimum segment size is 5 years, and the trend is calculated by linear regression.**

[Figure]

**Figure 13: PDF of the Tmax for the Blocking regime. The dotted black line represents the past period from 1979 to 2002, and the solid red line reflects the current period from 2003 to 2017. The purple vertical segment represents the threshold of the SU (Summer Days), i.e., 25°C. The blue vertical segment represents the temperature at which a frequency inversion occurs between the past period, with a higher frequency of temperatures between 25 and 30°C (blue coloured zone), and the current period, with a higher frequency of temperatures above 30°C (zone coloured orange).**

|  | Dynamical contribution [°C (*%*)] | Thermodynamic al contribution [°C (*%*)] | Residual term [°C (*%*)] | $\Delta T$ [°C] |
|---|---|---|---|---|
| WINTER (DJF) | 0.06 (*29.6*) | 0.17 (*78.9*) | -0.02 (-*8.5*) | 0.21 |
| SUMMER (JJA) | -0.05 (-*5.9*) | 0.87 (*103.2*) | 0.02 (*2.7*) | 0.84 |

**Table 3: Dynamical, thermodynamical and residual contributions of the temperature change ($\overline{\Delta T}$) in °C in winter (DJF) and in summer (JJA). Values in parenthesis give the ratio (in %) between the change components and the total change.**

**Commenté [JR15]: RC1-1:** New table with residual term

**RC3-7:** We delete the lines between winter and summer

[revised manuscript text omitted]